

# The Earthquake Risk Model of Switzerland ERM-CH23

Athanasios N. Papadopoulos [1], Philippe Roth[1], Laurentiu Danciu[1], Paolo Bergamo[1], Francesco Panzera[1,5], Donat Fäh[1], Carlo Cauzzi[1], Blaise Duvernay[2], Alireza Khodaverdian[3], Pierino Lestuzzi[3], Ömer Odabaşi[4], Ettore Fagà[4], Paolo Bazzurro[4], Michèle Marti[1], Nadja Valenzuela[1], Irina Dallo[1], Nicolas Schmid[1], Philip Kästli[1], Florian Haslinger[1], Stefan Wiemer[1]

[1] Swiss Seismological Service (SED), ETH Zurich, Zurich, 8092, Switzerland
[2] Federal Office for the Environment (FOEN), Bern, Switzerland
[3] École Polytechnique Fédérale de Lausanne, Lausanne
[4] RED Risk Engineering + Development, Pavia, Italy
[5] Department of Biological, Geological and Environmental Sciences, University of Catania, Italy

*Correspondence to*: Athanasios N. Papadopoulos (th.papadopoulos@outlook.com)

**Abstract.** Understanding seismic risk at both the national and sub-national levels is essential for devising effective strategies and interventions aimed at its mitigation. The Earthquake Risk Model of Switzerland (ERM-CH23), released in early 2023, is the culmination of a multidisciplinary effort aiming to achieve, for the first time, a comprehensive assessment of the potential consequences of earthquakes on the Swiss building stock and population. Having been developed as a national model, ERM-CH23 relies on very high-resolution site-amplification and building exposure datasets, which distinguishes it from most regional models to-date. Several loss types are evaluated, ranging from structural/nonstructural and contents economic losses, to human losses, such as deaths, injuries and displaced population. In this paper, we offer a snapshot of ERM-CH23, summarize key details on the development of its components, highlight important results and provide comparisons with other models.

## 1 Introduction

Natural hazards can cause widespread damage, loss of life, and disruption to critical services such as water, power, and transportation. Catastrophe risk models can aid governments and other stakeholders in determining the potential impact of different perils, identifying high-risk areas, and prioritizing resources and investments for preparedness and mitigation. They can further inform emergency response plans, helping increase the resilience of communities to catastrophic events. As such, the development and operation of national catastrophe risk models and related byproducts is increasingly seen as the basis for designing an effective data-driven disaster risk reduction strategy.

Strong earthquakes in particular, compared to other natural hazards, are characterized by rather infrequent occurrence and high potential for causing significant devastation. This low-probability-high-consequence feature of seismic risk hinders societal preparedness, as both public interest and actionable data are missing. In contrast, it makes modelling efforts all the more important for anticipating future scenarios and drawing mitigation actions. The latter could involve reduction (e.g., retrofit of



existing structures, update of design codes for new construction, educational campaigns), transfer (e.g., insurance) or planned retention (e.g., dedicated disaster funds) of the risk.

Seismic risk modelling initiatives at the national level have been undertaken to varying extents in different countries, such as
Italy (Dolce et al., 2021), USA (FEMA, 2010; Jaiswal et al., 2015), Canada (Hobbs et al., 2023), Germany (Tyagunov et al., 2006), Spain (Salgado-Gálvez et al., 2015), and Portugal (Marques et al., 2018). There have also been attempts to model risk at continental or global scales. In Europe, an open earthquake risk model (ESRM20; Crowley et al., 2021) has been released as an output of the European Union's Horizon 2020 SERA project (www.sera-eu.org). ESRM20 is a uniform risk model that covers 45 European countries and is the result of a concerted effort among the research community in Europe. Lastly, global
models have also been compiled, such as the UNISDR's Global Assessment Report (GAR; Cardona et al., 2014), and most notably, the 2018 Global Seismic Risk Map (Silva et al., 2020a) developed by the Global Earthquake Model (GEM) Foundation. The latter has allowed access to a uniform view of risk across the globe, a valuable resource, particularly for previously under-studied regions.

In Switzerland, the Federal Council commissioned in 2013 the Federal Office for the Environment (FOEN), in cooperation
with the Swiss Seismological Service (SED) and the Federal Office for Civil Protection (FOCP), to prepare a feasibility study and project plan to develop a national earthquake risk model. Based on these documents, the Federal Council commissioned in 2017 the SED, in cooperation with FOEN and FOCP, to develop this model until 2023. In the following sections, we give an overview of ERM-CH23 and its subcomponents and present primary results and insights.

## 2 Seismicity in Switzerland

In Switzerland, earthquakes are considered to be the natural hazard with the potential for causing the greatest damage. The 2020 Risk Report (FOCP, 2020), published by the FOCP, ranked earthquakes as the third largest risk faced by Switzerland, after electricity shortages and pandemics. Overall, the seismicity in the country is considered moderate with three to four earthquakes a day recorded on average within the country and around its borders by the Swiss Seismological Service (SED) at ETH Zurich. A destructive earthquake with moment magnitude ($M_w$) of 6.0 or above can be expected to occur on average
every 50 to 150 years (Wiemer et al., 2016). The most seismically active regions are found in the Valais and Graubünden cantons, as well as the southern Rhine Graben, a rift system located in the northeastern part of the country. The 1356 Basel earthquake is the largest known to-date earthquake to have struck Switzerland, with an estimated moment magnitude of 6.6. It caused widespread damage (Fäh et al., 2009) throughout Switzerland and neighbouring countries, and was felt as far away as Paris. Other notable historical events in the last 200 years include the 1855 $M_w$ 6.2 Visp earthquake (Fritsche et al., 2006) and
the 1946 $M_w$ 5.8 Sierre earthquake (Fritsche and Fäh, 2009).



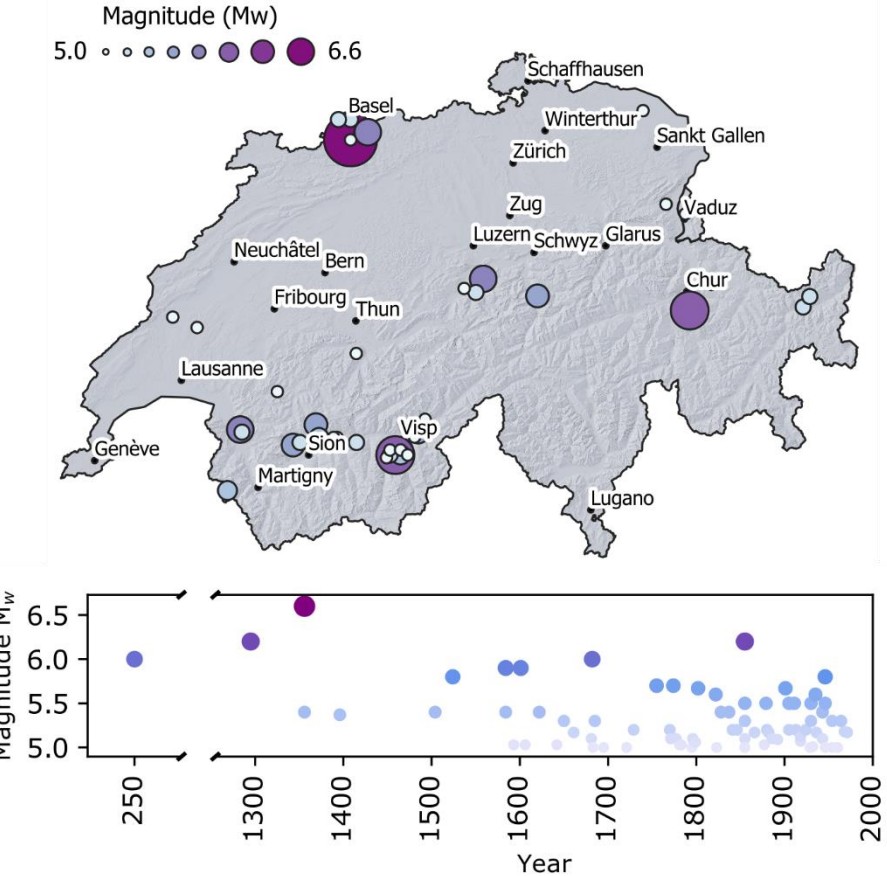

**Figure 1. Locations, dates and size of known historical earthquakes with magnitude $M_w$>5.0 in Switzerland (ECOS-09 catalogue; Fäh et al., 2011)**

## 3 The National Earthquake Risk Model of Switzerland (ERM-CH23)

### 3.1 Seismic hazard

ERM-CH23 primarily relies on the 2015 Seismic Hazard Model (SUIhaz2015; Wiemer et al., 2016), which is the authoritative national seismic hazard model of Switzerland. While SUIhaz2015 serves as the basis, a number of adjustments were made to tailor it for use within the context of ERM-CH23 and are succinctly detailed in the sections to follow, and in Wiemer et al. (2023) in more detail.

### 3.1.1 Source model

The seismogenic source model of SUIhaz2015 results from a weighted ensemble of four individual source models. The latter include an area source model (SEIS-15) and a smoothed seismicity model (CH14), developed specifically for SUIhaz2015.





The other two models are sourced from past seismic hazard models, i.e. SEIS04 (Wiemer et al., 2009) and ESHM13 (SHARE Project; Woessner et al., 2015). From the ensemble earthquake rate model, the activity rates corresponding to the 2.5%, 16%, 50%, 84%, 97.5% quantiles were obtained and assigned as five alternative logic tree branches. For further details on the source
model of SUIhaz2015, the reader is referred to Wiemer et al. (2016).

In ERM-CH23, the five original logic tree branches are "collapsed" into a single branch with weighted average rates. The motivation behind this choice was 1) to reduce the significant computational cost associated with risk analyses, and 2) also to avoid the synchronous assignment of improbable rates in all sources (e.g. in the 2.5th or 97.5th quantile rate branches) across the country. The bias from the later would invalidate any estimation of epistemic quantiles (although an argument exists that
it could be advantageous for the evaluation of mean estimates). Similar reasoning for using a single collapsed branch has been made in other models (e.g. Crowley et al., 2021).

The maximum magnitude is spatially variable and in the range of $M_w$ 6.5 to 7.3 (Wiemer et al., 2016). The minimum magnitude, originally set to $M_w$ 4.0 in SUIhaz2015, was increased to $M_w$ 4.5 for ERM-CH23, on the basis that smaller events are not of particular engineering significance (Bommer and Crowley, 2017).

**3.1.2 Ground shaking**

ERM-CH23 is built upon two main sub-models, one that uses spectral acceleration, henceforth referred to as SAM and given a weight of 0.7 in the overall logic tree (Figure 6), and one that uses EMS-98 (Grünthal, 1998) macroseismic intensity, henceforth referred to as MIM and given a weight of 0.3. The ground shaking in SAM is modelled with the same set of ground motion models (GMMs) used in SUIhaz2015 (Wiemer et al., 2016; Edwards et al., 2016). These include empirical models
based on datasets in Europe and worldwide, such as those of Zhao et al. (2006) Chiou and Youngs (2008), Cauzzi and Faccioli (2008), and Akkar and Bommer (2010), adjusted to match the amplification and attenuation levels typical of the Swiss reference rock (Edwards et al., 2016; Wiemer et al., 2016; Poggi et al., 2011)(Wiemer et al., 2016). They also include the Swiss-specific stochastic models of Edwards and Fäh (2013) and Cauzzi et al. (2015), obtained by simulating ground shaking for various source, path and site-specific parameterizations. Different GMMs and weights are set for each of four identified
tectonic regimes, namely Alpine Shallow, Alpine Deep, Foreland Shallow, Foreland Deep. Each tectonic regime represents a different branching set in the logic tree, with 18, 16, 18 and 16 GMMs, respectively. The total number of GMM logic tree branch combinations reaches 82,944 (18 x 16 x 18 x 16). Further details on the selection, weighting, and statistical performance of these models in Switzerland can been sought in Edwards et al. (2016).

For what concerns MIM, a selection of intensity prediction equations (IPEs) was carried out for ERM-CH23. A residual
analysis was conducted on the macroseismic dataset for the region, in order to compare a collection of candidate IPEs. The latter were ranked and four of them (Table 1) were then selected to represent the body, center and range of intensity data.

**Table 1. Intensity prediction equations used in ERM-CH23**



| Name | Weight | Reference |
|------|--------|-----------|
| ECOS09variableDepth | 0.2 | (Fäh et al., 2011) |
| ECOS09fixedDepth | 0.3 | (Fäh et al., 2011) |
| BaumontEtAl2018High2210IAVGDC30n7 | 0.3 | (Baumont et al., 2018) |
| Bindi2011RHypo (with conversion to moment magnitude $M_w$) | 0.2 | (Bindi et al., 2011) |

Finally, some adjustments were carried out in the aleatory uncertainty modelling of the IPEs and GMMs. For the former, it was decided to use the intra- and inter-event sigma of the Baumont et al. (2018) model, since the other functions do not distinguish into intra- and inter-event components, which is important for risk analyses. On the GMM side, the inter-event sigma of the original functions is maintained, while the intra-event sigma is modelled as site-specific and derived together with the site amplification model (see following section) to ensure compatibility.

### 110 3.1.3 Site amplification

As a component of ERM-CH23, a new ground motion site amplification model was developed (Bergamo et al., under review; Wiemer et al., 2023), covering the entire Switzerland in a homogeneous manner. This model is based on two datasets. The first one comprises site amplification factors that were measured at seismic stations across Switzerland, while the second one is composed of site condition indicators that are known to be correlated with local seismic response. The empirical spectral

modeling technique (ESM; Edwards et al., 2013), was used to compute Fourier amplification functions at instrumented sites, using earthquake recordings from 2000 to 2021; the amplification was then translated from the Fourier to the pseudo-spectral acceleration (Sa) domain resorting to random vibration theory (Liu and Pezeshk, 1999). The site condition indicators, including the lithological classification of Switzerland, multiscale topographic slope, and depth-to-bedrock, were combined with the empirical amplification factors using the regression-kriging algorithm (Hengl et al., 2007). To allow coherent integration of

the ground motion and site amplification modules, corresponding maps of the site-to-site variability ($\varphi_{S2S}$) and the single-site within-event variability ($\varphi_{SS}$) were also produced and used to define the overall intra-event variability. The latter site-specific estimate was used to replace the intra-event uncertainty term of the employed GMMs. The site model is derived for intensity measures, namely Peak Ground Velocity (PGV) and pseudo-spectral acceleration at 3 periods (0.3 s, 0.6 s and 1.0 s). The Sa(0.3 s) and Sa(0.6 s) models are employed in SAM, whereas the site amplification maps for PGV, Sa(0.3 s) and Sa(1.0 s)

were further translated into macroseismic intensity aggravation layers (e.g. Figure 2) for use with the IPEs and associated macroseismic intensity-based vulnerability functions. The conversion to macroseismic intensity aggravation was performed using the Faenza & Michelini (2011, 2010) relations and the correction factors estimated by Panzera et al. (2021), the latter to take into account the shift from the reference soil condition of the GMMs to that of the IPEs.





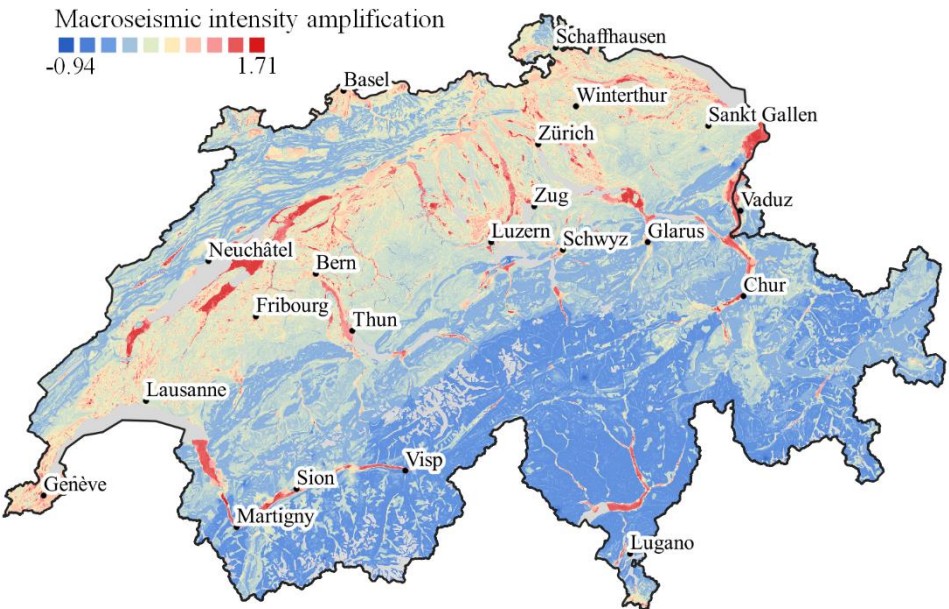

Figure 2. Example of macroseismic intensity aggravation map, derived from a Sa(0.3 s) proxy.

## 3.2 Seismic vulnerability

### 3.2.1 Taxonomy

According to several surveys carried out in Switzerland (e.g. Lestuzzi et al., 2016), the building taxonomy proposed in Lagomarsino and Giovinazzi (2006) is suitable for application to Switzerland with minor modifications (Table 2).

Table 2. ERM-CH23 building taxonomy

| Typology | Description | | Height | Typology | Description | Height |
|---|---|---|---|---|---|---|
| M1_L | Unreinforced | masonry | ≤ 3 stories | S | Steel | any |
| M1_M | (Dry stone) | | 4 – 6 stories | T | Timber | any |
| M3_L | | | ≤ 3 stories | M6_L | Unreinforced | ≤ 3 stories |
| M3_M | Unreinforced | masonry | 4 – 6 stories | M6_M | masonry – RC | 4 – 6 stories |
| M3_H | (Rubble stone) | | ≥ 7 stories | M6_H | floors | ≥ 7 stories |
| M4_L | | | ≤ 3 stories | RCmix_L | | ≤ 3 stories |
| M4_M | Unreinforced | masonry | 4 – 6 stories | RCmix_M | Mixed shear wall | 4 – 6 stories |
| M4_H | (Dressed stone) | | ≥ 7 stories | RCmix_H | and RC frame | ≥ 7 stories |
| M5_L | Unreinforced masonry (old | | ≤ 3 stories | RCW_L | | ≤ 3 stories |
| M5_M | bricks) | | 4 – 6 stories | RCW_M | Shear wall | 4 – 6 stories |



| M5_H | | ≥ 7 stories | RCW_H | ≥ 7 stories |
|------|---------------|-------------|-------|-------------|
| Ind | Industrial type | any | | |

### 3.2.2 Fragility functions

Two different sets of fragility curves were derived, one in terms of macroseismic intensity for the MIM logic tree branches of the overall model and one in terms of spectral acceleration (at 0.3s or 0.6s) for the SAM branches (Wiemer et al., 2023). The

MIM fragility model relies on the methodology described in Lagomarsino and Giovinazzi (2006), Lagomarsino et al. (2021), and Bernardini et al. (2010), together with engineering judgment about Swiss practice.

For the development of the SAM fragility functions, a statistical investigation of building blueprints was first exploited to identify average geometric characteristics of various building types. Capacity curves, idealized in bilinear form, were then obtained from numerical models (Lestuzzi et al., 2017). For most typologies 1000 capacity curves were then generated using

the statistical model, covering material uncertainties. The method detailed in Michel et al. (2018) was followed to derive analytical fragility curves (Wiemer et al., 2023).

### 3.2.3 Consequence model

A consequence model that relates damage to loss has been compiled for application to Switzerland (Wiemer et al., 2023). Different approaches were used for each of the five loss types of interest, depending on the availability of data. In brief, injuries

and deaths were modelled based on the estimates given by HAZUS (FEMA, 2010), NCPD (2018), and Spence et al. (2007). Estimates of displaced population were instead adopted from the empirical data harmonized by the Italian National Civil Protection Department (NCPD, 2018). Displaced population in ERM-CH23 refers to households that have been displaced either in the short- or long-term. Content damage-to-loss estimates have also been adopted from the literature, and more precisely from HAZUS (FEMA, 2010).

On the other hand, the structural/nonstructural damage-to-loss functions have been derived analytically adopting the loss estimation methodology of FEMA P-58 (FEMA, 2018). For each building typology, the prescriptive damage states as per the EMS-98 scale were matched to associated structural demand thresholds sourced from the literature (Wiemer et al., 2023). Archetype blueprints were used to infer quantities and features of structural elements such as load-bearing masonry walls, spandrels and slabs. The quantity estimator tool of FEMA P-58 was also used to determine the non-structural component

quantities with uncertainty. Fragility and consequence functions for damageable structural and non-structural components, present in Swiss buildings, that were not available in FEMA P-58, were gathered and collated from other sources. The repair and replacement costs were adjusted using a macro-economic model in view of the construction dynamics between the reference country (from which cost functions were available) and Switzerland. The development of the structural/nonstructural damage-to-loss model is further described in Wiemer et al. (2023).





### 3.2.4 Vulnerability functions

Vulnerability functions were derived by combination of the fragility and consequence models. Figure 3 shows a comparison of the structural/nonstructural vulnerability curves for the four most prevalent typologies in Switzerland (see Figure 5). As expected, the pure masonry typologies (M3, M5) are more vulnerable, with damage onset expected from rather low intensity levels. The reinforced concrete wall (RCW) buildings, as well as the M6 typology which combines concrete and masonry elements are thought to be less vulnerable with damage expected at higher intensity values.

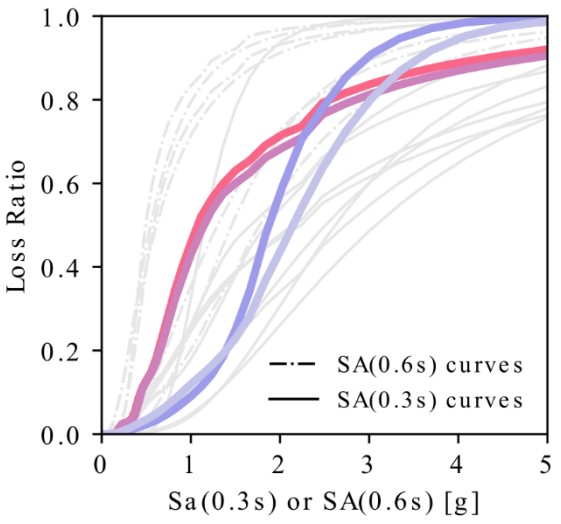
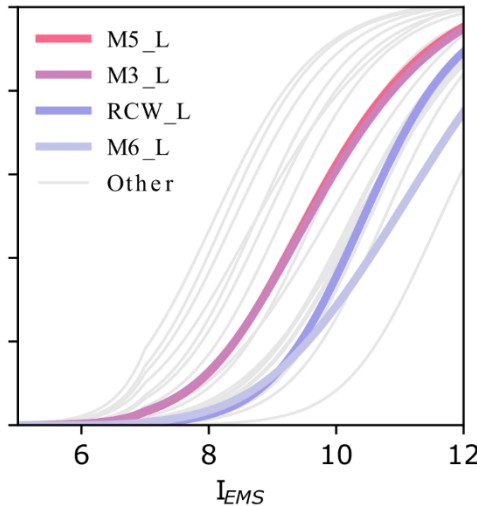

**Figure 3. SAM (left) and MIM (right) structural/nonstructural vulnerability curves**

### 3.3 Building exposure

The exposure model describes the location, value, occupants and typological characteristics of the buildings at risk. At the base of the ERM-CH23 exposure model lies an extensive geo-referenced database of all building objects in Switzerland, assembled by the Federal Office for the Environment (FOEN). Further details on the building database can be found in FOEN (2021) and Hügli et al. (2021), as well as in Wiemer et al. (2023). ERM-CH23 makes use of over 2.25 million building entries in the database, after excluding close to 900'000 objects with an unclassified function or a volume above ground smaller than 200 m$^3$ (typically bungalows, storage sheds, bus shelters, etc).

Among others, the building database includes information such as the period of construction, building function, footprint area, volume and height (which is then used to define the number of stories) as obtained from digital surface and digital terrain elevation models. The reconstruction (replacement) cost is determined for each building according to Röthlisberger et al. (2018), using the building volume, the building function and the building zone category as predictor variables. The replacement value of building contents is computed as a fraction of the building reconstruction value, and varies from 0.19 to 0.65 depending on the building function. The modelled values were further validated using data from the cantonal building insurance



companies. The number of occupants in each building is defined through de-aggregation of geo-referenced housing and employment statistics. For the estimation of human losses, ERM-CH23 uses a static (time-agnostic) equivalent number of occupants in each building that is obtained as a weighted average of residents, employees, students (in school buildings) and patients (in hospitals), as well as allowing for a share of the population being outdoors at the time of the earthquake. For further
details the reader is referred to Wiemer et al. (2023).

Most of the exposure is concentrated at the Swiss plateau, north of the Alps (Figure 4a), especially around urban centres, such as Zurich, Geneva, Basel and Bern. The vast majority of the Swiss building stock consists of low-rise buildings of 1 to 3 stories (Figure 4b). High-rise buildings (>7 stories) are quite rare and concentrated in the major urban centres, such as Geneva, Zurich and Basel. Figure 4d shows the distribution of buildings constructed in different time periods. It appears that only a small
fraction of the total was built after the introduction of seismic codes in 2003, with a significant amount of construction having taken place within the 1971-1990, 1946-1970 and <1919 periods. The total value of the modelled building stock and contents amounts to 2.9 trillion CHF and 0.8 trillion CHF, respectively. About 70% (structural/nonstructural) and 54% (contents) of this value comes from residential buildings. Commercial and public buildings add up to about 18% and 23% of the total structural/nonstructural and content values, while industrial buildings add up to about 10% and 20%, respectively. Agricultural
buildings make up about 3% of the total value. The share of buildings of different occupancy by canton is illustrated in -Figure 4c.



**Figure 4. (a) Spatial distribution of buildings across Switzerland, (b) distribution of buildings by number of stories, (c) cantonal distributions of buildings by occupancy, (d) distribution of buildings by construction period.**

The prevalence of different structural typologies in the Swiss building stock was assessed by means of field surveys carried out in the cities of Basel, Solothurn, Sion, Yverdon-les-Bains, Neuchâtel and Martigny (e.g. see Diana et al., 2019). Subsets of the building stock at these locations were visually assessed and assigned to a structural typology as per the taxonomy given in Table 2. The survey outcomes were used in two alternative ways within ERM-CH23 (Wiemer et al., 2023). In the first one, the statistics of structural systems were obtained, conditional on two attributes: the construction period and the height of the

building (1-3 stories, 4-6 stories, >7 stories). Subsequently, a structural typology was assigned to each database entry by random sampling from the aforementioned conditional statistics. The second approach involved training a random forest algorithm (Tin Kam Ho, 1998) on the attributes of the database matched to the results of the field surveys. The random forest algorithm was then executed to predict the typologies of the remaining database entries. The two approaches constitute two



alternative logic tree branches of the ERM-CH23 framework. The rate-based (RB) approach was given a weight of 0.25 and
the random forest (RF) based procedure was given a weight of 0.75. Figure 5 shows the share of the main typologies in
Switzerland, as well as in the large municipalities of the main urban centers and in the smaller municipalities comprising the
rest of the country. In general, masonry is the predominant construction material, with M3 (unreinforced rubble stone masonry),
M5 (unreinforced old brick masonry), and M6 (unreinforced masonry with RC floors) being the most common. Reinforced
concrete wall (RCW) buildings also make up a significant, albeit lower, fraction of the building stock.

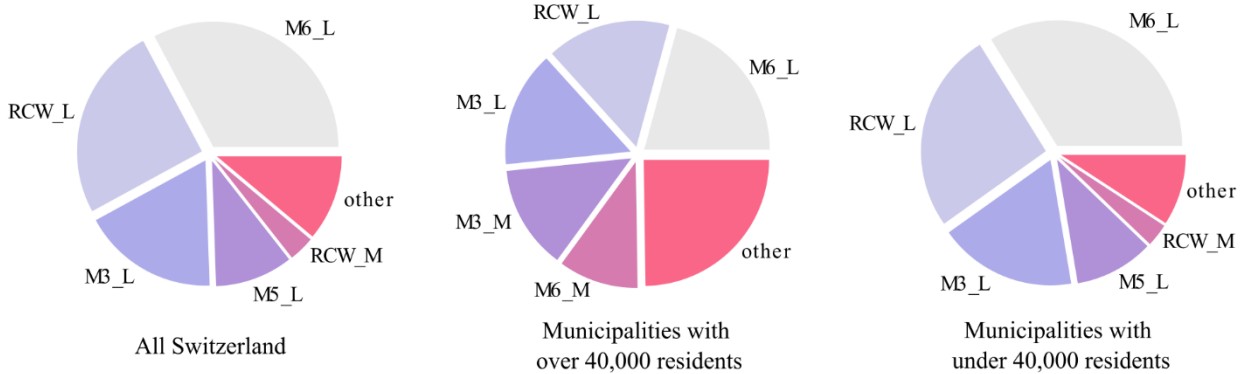

**Figure 5. Share of different building typologies in the entire Swiss building stock (left), in large municipalities (middle), and in small municipalities (right)**

Since the ERM-CH23 exposure numbered more than 2 million individual buildings, it had to be aggregated on a spatial grid
to facilitate the risk computation. After investigating different options (Wiemer et al., 2023, Papadopoulos et al., in
preparation), the aggregation was performed on a 2 km x 2 km regular grid, along with some further considerations for
minimizing any resulting errors. More precisely, the site parameters at the locations of buildings within each cells were first
clustered with the K-means (MacQueen, 1967) approach. Buildings (of the same typology and postal code) belonging to each
of five (SAM) or three (MIM) grid cell clusters were placed in adjacent locations near the cell centroid and merged into one
macro-asset. At each cluster location, the associated site parameters were assigned (Papadopoulos et al., under review).
Moreover, the merging of buildings into macro-assets (i.e., single assets with replacement values equal to the sum of the values
of the buildings being aggregated), implies a perfect correlation of the ground motion and loss residuals (given ground motion)
across the buildings being aggregated. To remove the effect of this implicit correlation of the loss ratios, vulnerability curves
for macro-assets of $n$ buildings (where $n$ was taken equal to 1, 5, 20, and 85) were estimated and used for macro-assets of
different sizes. This was done by sampling the single building loss ratio multiple times for each of the $n$ buildings comprising
it, summing up to get the total macro-asset loss, and then building the updated loss ratio distribution given each ground motion
level (Wiemer et al., 2023, Papadopoulos et al., in preparation).





## 3.4 Modelling of uncertainty

Undoubtedly, there are large uncertainties involved in earthquake risk modelling. As usual, aleatory uncertainties are
considered in the modelling of earthquake occurrence (via a Poisson process), in the modelling of ground motion, as well as
in the modelling of loss given ground motion. Epistemic uncertainties in ERM-CH23 are captured via a logic tree approach,
as already alluded earlier on. Figure 6 illustrates the logic tree set up that was adopted. A primary branching distinguishes the
MIM and SAM sub-models, with further branching levels for ground shaking modelling, site amplification and building
mapping. The logic tree numbers 24 MIM-specific end-to-end branches and 165,888 SAM-specific end-to-end branches. For
the risk calculations, all MIM branches were considered, whereas 400 SAM branches were randomly (based on their weights)
selected and analysed.

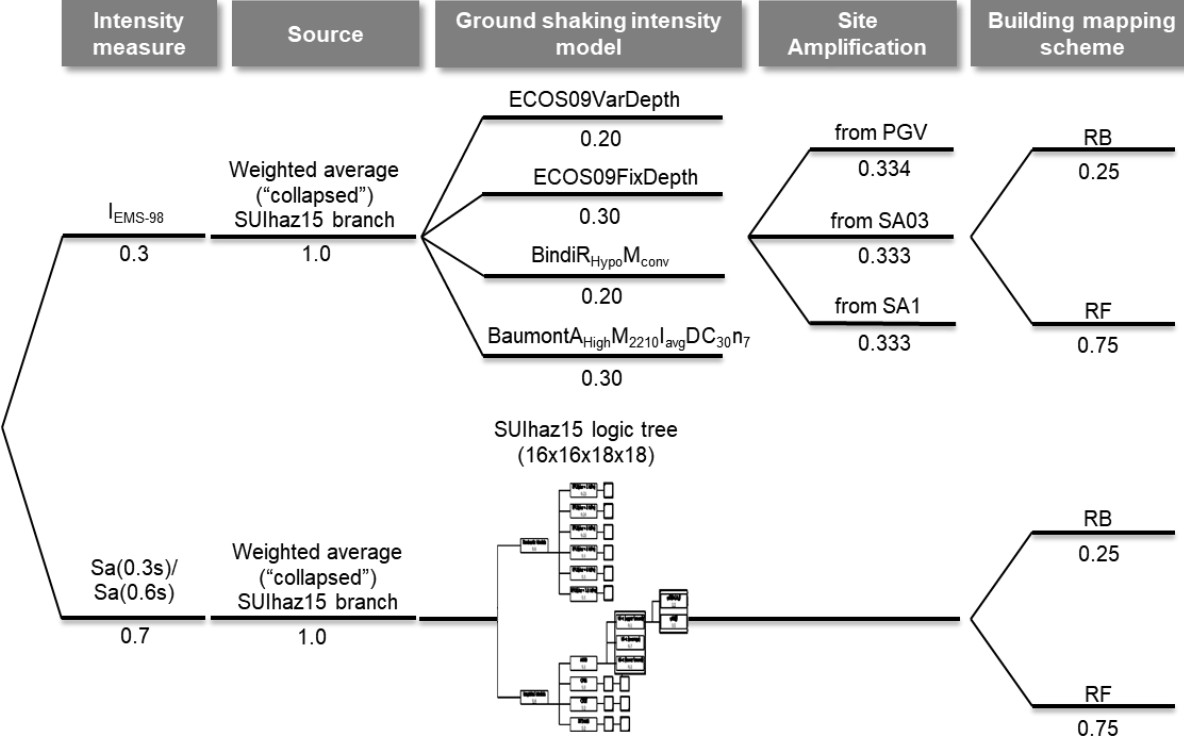

**Figure 6. The logic tree of ERM-CH23**

## 4 Earthquake risk assessment

To assess the earthquake risk over a spatially distributed exposure, a so-called event-based approach based on Monte-Carlo
simulations is typically required. An event-based probabilistic earthquake risk assessment starts with the generation of a large
number of stochastic earthquake catalogues and the generation of associated random ground motion fields for each rupture in
the catalogue. The simulated ground motion intensity values at each site are then passed onto the vulnerability functions





associated with the building typologies at each site. The asset-specific losses are then sampled and added to compute the total
loss for the given earthquake. Finally, the sample of loss estimates is then used to obtain standard risk metrics, such as average
annual losses (AAL) and probable maximum loss (PML) curves.

All calculations are carried out using the open-source OpenQuake engine v3.14 (Pagani et al., 2014) developed by the GEM
foundation. For each of the 24 MIM logic tree branches, 20,000 1-year long stochastic earthquake catalogues were generated,
while 10,000 1-year long catalogues were simulated for each of the 400 SAM logic tree sampled branches. In total, this resulted
in 4.48 million 1-year-long stochastic catalogues, a number that was deemed sufficient to achieve acceptable convergence for
the quantities of interest.

Figure 7 presents the obtained AAL (epistemic) distributions for four of the five loss types that were considered, while Figure
8 shows the obtained PML curves for structural/nonstructural economic loss and fatalities. ERM-CH23 predicts a direct
economic AAL of 245 M CHF (or 0.084‰ of the total value) from structural/nonstructural components, plus another 28 M
265 CHF (or 0.033‰ of the total value) from contents. This annual economic loss amounts to about 0.03-0.04% of Switzerland's
gross domestic product (GDP). The AAL for fatalities, injuries and displaced population is estimated equal to 7.6, 59.5, and
1079.7, respectively. Structural/nonstructural losses of around 10 billion CHF and about 300 fatalities are expected to be
exceeded every 200 years on average. Likewise, for a 1000 year return period of exceedance the loss estimates are assessed at
around 37 billion CHF and close to 1700 fatalities.



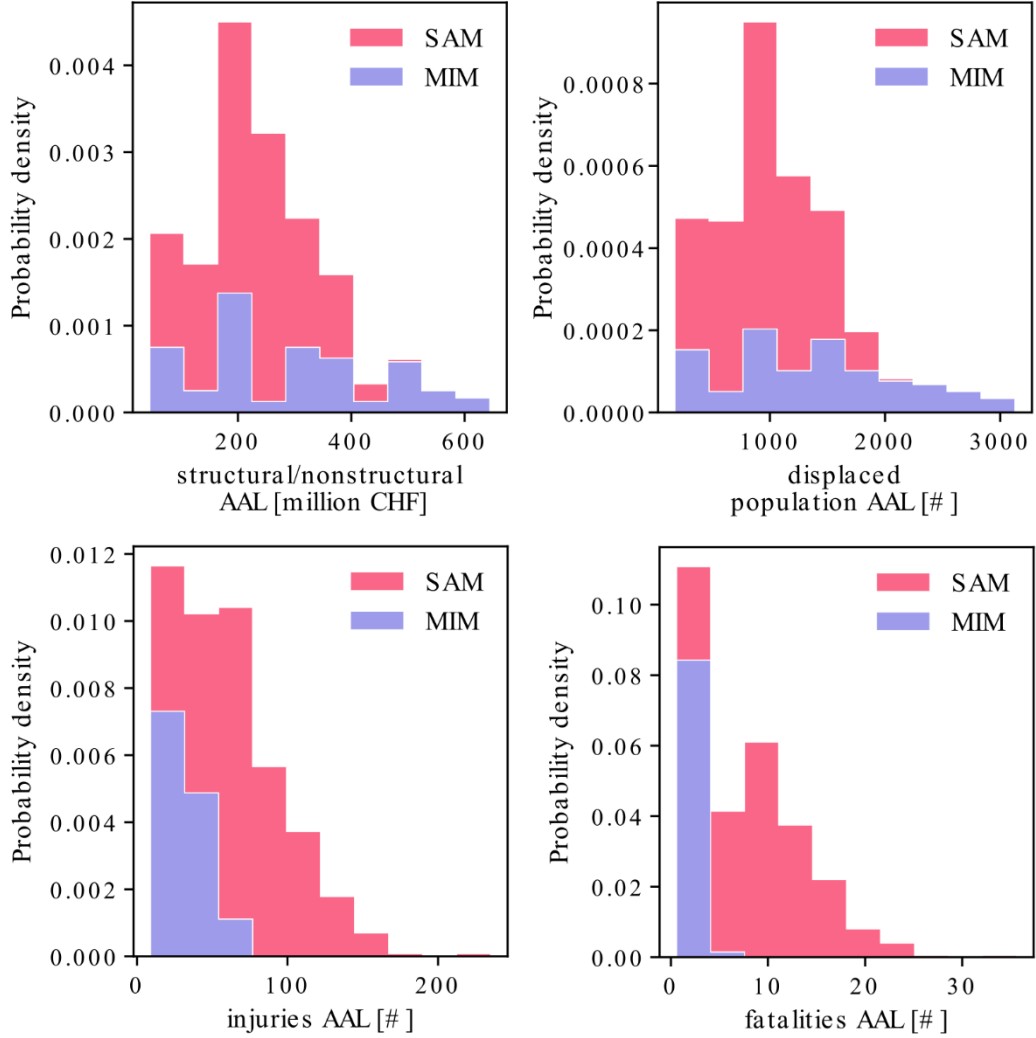

270

**Figure 7. Average annual loss (AAL) epistemic distributions for four loss types. The contributions of the MIM and SAM models are stacked, i.e. at each x-axis bin, the relative heights of the SAM and MIM bars indicate the contribution of the two models at the particular loss value.**



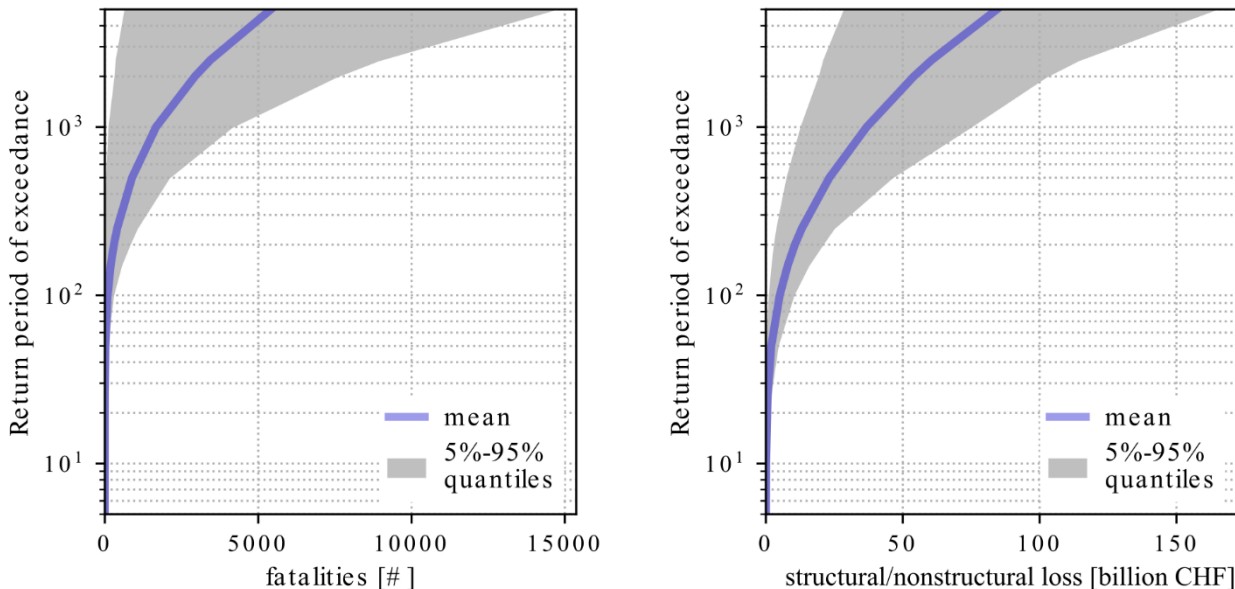

**Figure 8. Probable maximum loss curves distributions for fatalities (left) and structural/nonstructural economic loss (right).**

Of course, as shown in both Figure 7 and Figure 8, there is non-negligible dispersion around the mean estimates reported above, which reflects the large uncertainties in many parts of the model. The main driver of the epistemic uncertainty is the modelling of ground shaking as indicated by the tornado diagrams (Porter et al., 2002) in Figure 9, an observation that is in line with previous studies (e.g. Field et al., 2020). For structural/non-structural AAL, the choice of IPE/GMM leads to a ~5-fold difference, whereas for fatality AAL the difference is ~4-fold for IPEs and 35-fold for GMMs. Important differences are also observed between the two submodels, MIM and SAM, especially for fatalities. Lastly, the building mapping scheme and site amplification uncertainties explain a smaller part of the total uncertainty around the country-wide AAL. That said, note that even the latter two sources of epistemic uncertainty might lead to significant differences at local scales (see Wiemer et al., 2023), making their inclusion in the model very important.



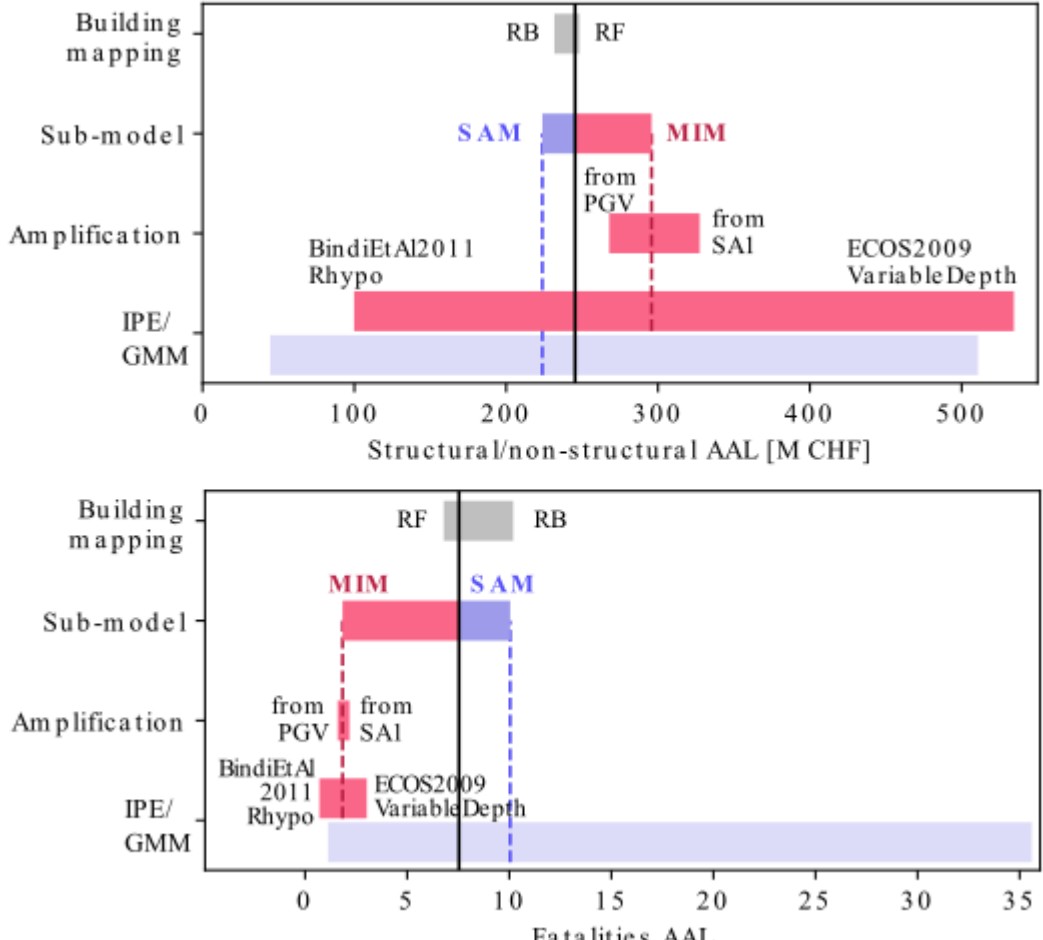

**Figure 9. Epistemic variable tornado diagrams for structural/non-structural (top) and fatality (bottom) AAL. The bars show the minimum and maximum AAL estimate if only the most extreme branches were used at each level, while the remaining logic tree remained the same. The MIM and SAM specific bars refer to estimates of those sub-models rather than of the entire model. Finally, in the case of GMMs, since enumeration is not possible and 400 branches are sampled, the bars simply refer to the minimum and maximum values obtained across these 400 samples.**

Across the country, the highest AAL estimates are naturally found in areas that combine high concentration of exposure with elevated seismic hazard. The first panel in Figure 10 shows the breakdown of structural/nonstructural AAL across the Swiss cantons. Overall, populous cantons such as those of Bern (BE), Zurich (ZH), and Vaud (VD) feature some of the highest AAL estimates (largely) due their large building stock. High AAL estimates are also found for cantons such as Basel (BS) and Valais (VS) that combine higher seismic hazard with decently sized exposure. The second panel of Figure 10 presents the spatial distribution of AAL ratio (AALR) by municipality. Here, we see that when losses are normalized by the total replacement cost, the spatial pattern tracks the pattern of seismic hazard (on soil conditions). Indeed, municipalities in the south-western canton of Valais stand out, as a result of the increased seismicity rates and high site amplification along the valley, where most cities are located.





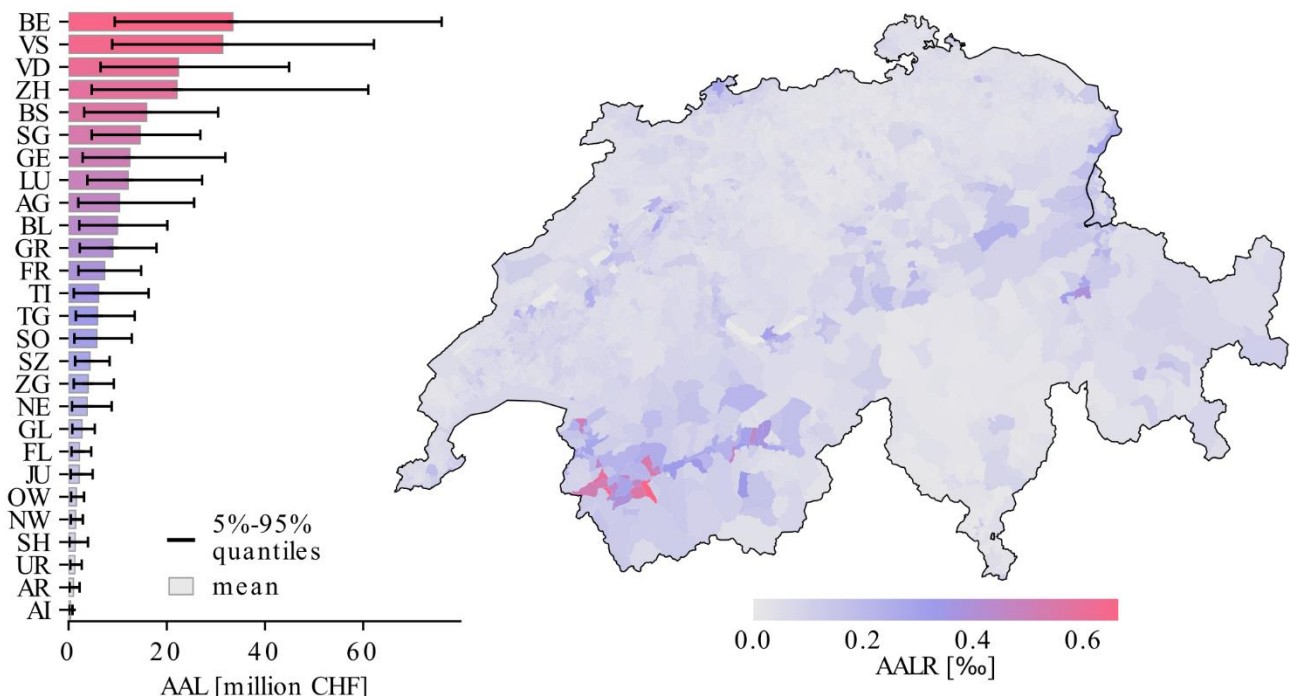

**Figure 10. Structural/nonstructural AAL by canton (left) and AALR by municipality (right)**

Figure 11 compares the structural/nonstructural AAL and AALR obtained for different structural typologies. Overall, M1, M3 and M5 typologies, and especially the mid- and high-rise variations, display the largest AALR. However, the largest contributions towards the total country-wide AAL come from the M3_L, M6_L, M3_M and RCW_L classes. This reflects the combination of their frequency within the exposure model and their relative vulnerability.



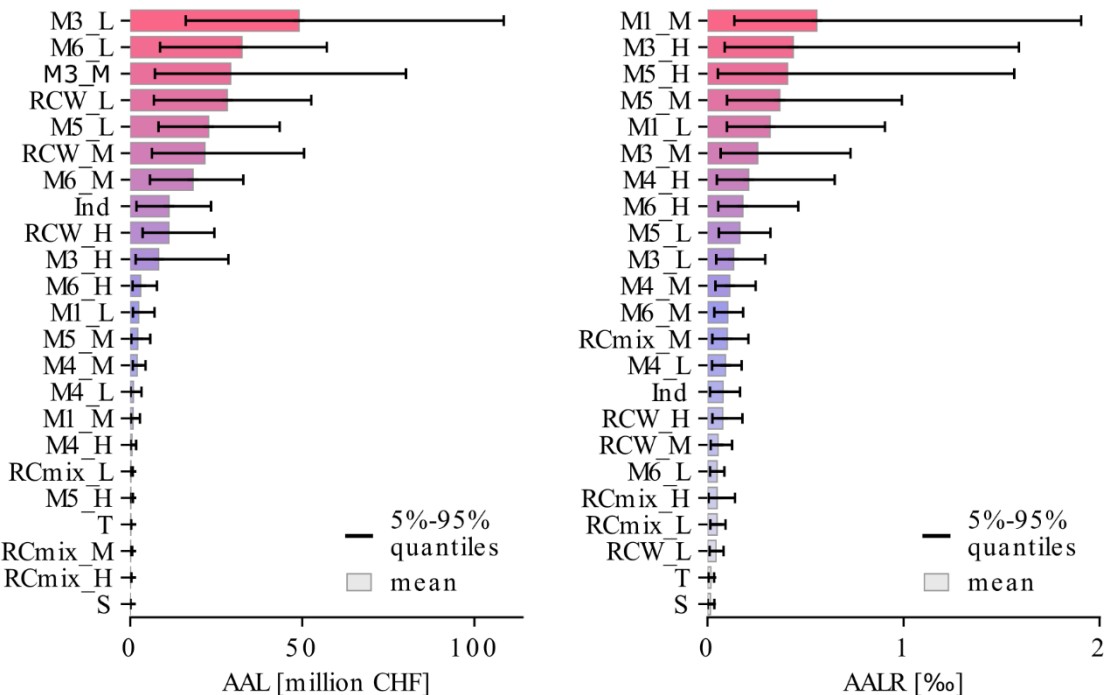

**Figure 11. Structural/nonstructural AAL (left) and AALR (right) by building typology**

## 5 Discussion

The risk view provided by ERM-CH23 offers a basis for earthquake risk management in Switzerland. That said, there are several dimensions of earthquake risk that ERM-CH23 does not cover in its first iteration. Earthquake losses quantified by the model refer, solely, to loss induced to the building stock by direct physical damage caused by ground shaking. Damage to infrastructure services (water, wastewater, energy, telecommunications, transport, etc.) and secondary effects (Daniell et al., 2017) from soil liquefaction, landslides, lake tsunamis, fire following or triggered technological accidents (Na-Tech) are not modelled. Indirect losses, such as those incurred from business interruption are also not estimated, nor are possible demand surge effects due to scarcity of human and material resources and overall disruption of supply chains.

An important hurdle for modelling risk in areas of low and moderate seismicity is the lack of historical data for model calibration and validation. Lacking past damage observations, validation was based on subjecting the individual components, as well as the overall model, to sanity checks and verification exercises. Damage and loss analyses were carried out for a wide range of earthquake scenarios. The spatial pattern of modelled ground motion and damage from small and large earthquakes was qualitatively assessed and contrasted with other models and observations. The relative vulnerability of the considered building classes was also subjected to scrutiny and it was made sure that it reasonably matches engineering expectations. The





development of the model was followed by a panel of independent experts to ensure that it conforms with current state-of-the-

art practices, while the finalized model was presented and received peer-review by a second independent expert panel.

Lastly, comparisons with the recent European Seismic Risk Model (ESRM20; Crowley et al., 2021) model, GEM's 2018 Global Risk Model (Silva et al., 2020b), and the 2015 Global Assessment Report (GAR; Cardona et al., 2014) were conducted to place our results among other estimates and understand the reasons for any deviations. Table 3 compares AAL estimates between the aforementioned models and ERM-CH23, while Figure 12 contrasts their reported PML curves. GAR15, which is

the least detailed model of the four, yields the higher loss estimates with a frequency that seems generally on the high-side. ESRM20 and GEM18 on the other hand predict significantly lower losses compared to ERM-CH23. The difference is smaller when looking at the AALR, which indicates a large difference in the total exposure value considered between the models. This difference can explain about half the discrepancy with ESRM20 and the most part of the discrepancy with GEM18. ERM-CH23 uses a near-complete database of building objects within the country, whereas the assessment of replacement costs is

informed by cantonal insurance sources, which lends credibility to the modelling. A second observation noteworthy is the increased granularity of the site amplification modelling in ERM-CH23. Comparisons with ESRM20 indicated a higher range of site amplification factors in ERM-CH23. This meant higher site amplification in several areas with soft soil deposits (usually around lakes and rivers, where many cities and settlements are located) and lower in mountainous areas with scarce exposure. This can explain further differences between ESRM20 and ERM-CH23. Of course, pinpointing the exact factors behind model

differences is challenging, since all these models employ very different ground motion, exposure and vulnerability components.

**Table 3. Comparison of ERM-CH23 AAL estimates with other models**

|  |  | ERM-CH23 | ESRM20 | GEM18 | GAR15 |
|---|---|---|---|---|---|
| Structural/Non-structural loss | AAL | 245 M CHF | 55 M EUR | 100 M USD | 785 M USD |
|  | AALR [‰] | 0.084 | 0.043 | 0.07 |  |
| Fatalities | AAL | 7.6 | 2 |  |  |
|  | AALR [‰] | 0.00099 | 0.0002 |  |  |

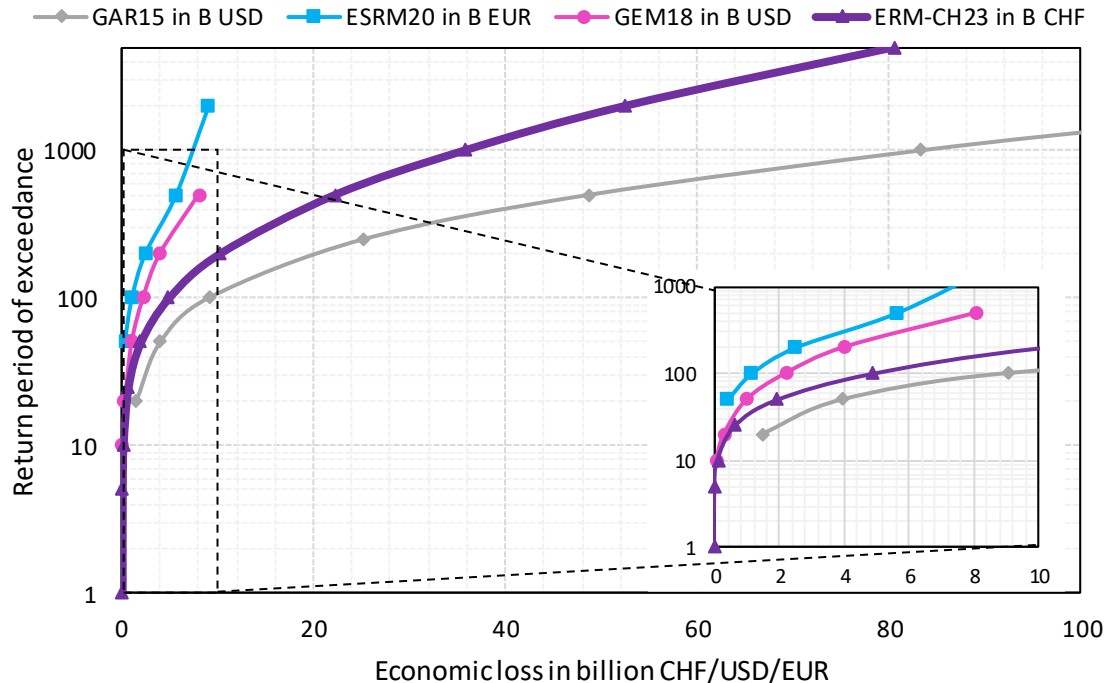

**Figure 12. Comparison of structural/nonstructural PML with other models**

## 6 Communication products and user testing

A dedicated communication concept was needed to bring the insights of the first publicly available earthquake risk model to our different target audiences (Bentele and Nothhaft, 2007). This communication concept consisted of (Wiemer et al., 2023): i) a SWOT (Strengths, Weaknesses, Opportunities, and Threats) analysis to identify the strengths, weaknesses, opportunities, and threats; ii) a definition of the target audiences; iii) a list of the communication goals and key messages; iv) a description of the communication products; v) a planning of the (release) events; and vi) a strategy for the user testing.

Based on the target audiences' needs we developed various products. Some of them support decision-making for earthquake preparedness and response (e.g., scenarios and rapid impact assessments), and others inform about the seismic risk model and its results (e.g., flyer, poster, explainer video, technical report). A key product is the earthquake risk map which depicts an index that combines the expected number of fatalities with the estimated financial losses due to building damage (Figure 13). Further, we developed an earthquake risk tool which allows interested people to determine by approximation their personal earthquake risk. The assessment of the personal earthquake risk is based on three factors at the indicated location: the earthquake hazard, the local amplification, the vulnerability of a building depending on the number of storeys and construction



period. All products are available on the SED website in the three national languages of Switzerland and in English (see

seismo.ethz.ch).

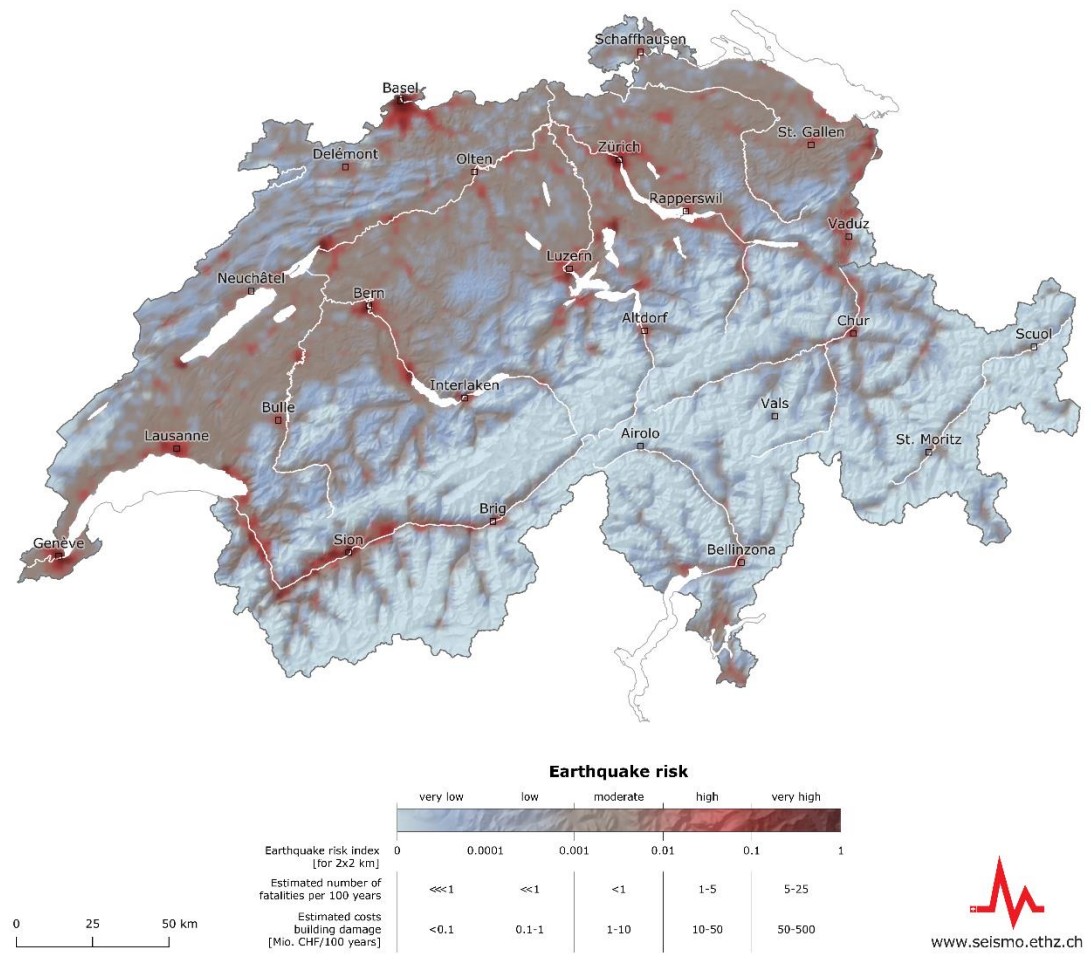

**Figure 13. The earthquake risk map of Switzerland**

To design these user-centered products, testing was indispensable (Dallo et al., 2022; Marti et al., 2019). We thus followed a
transdisciplinary approach since we – an interdisciplinary group consisting of model developers, IT specialists, and
communication experts – co-developed the products and tested them with the target audiences. We first conducted interviews
with international experts to learn from best practices and already operational seismic risk services (e.g. Pager; Dryhurst et al.,
2021). Second, we organised workshops with professional stakeholders (e.g., cantonal authorities, civil protection) to assess
their information and application needs, and to further develop the prototypes based on their feedback (Marti et al., 2023).
Third, we conducted two public surveys to evaluate which rapid impact assessments, scenarios, and risk maps are correctly

interpreted, perceived as useful, and preferred (Dallo et al., 2023; Marti et al., 2023). For the product design, we further benefitted from our experiences of the release of the first European Seismic Risk Model (ESRM2020; Dallo et al., in preparation).

## 375 7 Conclusion

This study summarized the development of the first Earthquake Risk Model of Switzerland, ERM-CH23, and provided key results. ERM-CH23 represents an important milestone in advancing the understanding of earthquake risk in the country. Estimates of the size and spatial pattern of earthquake risk in Switzerland was previously lacking from the public domain and inferences had to be made relying on solely hazard information. By filling this gap, our hope is that ERM-CH23 will encourage

evidence-based decision-making by public authorities and other stakeholders, in efforts towards risk mitigation and disaster resilience. Further downstream products of the ERM-CH23 project are also expected to underpin disaster preparedness and response. A rapid impact assessment service has also been devised, using the ERM-CH23 framework to produce near real-time estimates of damage and loss after the occurrence of earthquakes. This system will use ground motion footprints updated with station recordings, as implemented in the Swiss ShakeMap service (Cauzzi et al., 2015). In the future, the National

Earthquake Risk Model of Switzerland should be periodically updated and improved, incorporating the latest science and datasets. Extensions to cover secondary perils, indirect losses and infrastructure should also be planned to enable a holistic view or earthquake risk.

## 8 Author contribution

Conceptualization: ANP, DF, MM, LD, BD, PL, SW; Data curation: ANP, FP, BD, LD, AK; Formal analysis: ANP, FP, PB, DF, OO, LD, BD, AK,; Funding acquisition: SW, DF, PR, FH, BD; Methodology: ANP, FP, PB, DF, MM, ID, OO, EF, CC, AK, PL, LD, BD; Project administration: PR, DF, NV, MM, FH, SW; Resources: SW, MM, ID, PR, NS, PK, LD; Software: ANP, CC, NS, PK, LD; Supervision: SW, ANP, DF, PB, PR, LD, PL; Validation: ANP, DF, CC, PR, LD, BD ; Visualization: ANP; Writing: ANP, NV, ID; Writing – review & editing: ANP, FP, PB, DF, CC, PR

## 395 9 Competing interests

The authors declare that they have no conflict of interest.



## 10 Acknowledgements

The development of the ERM-CH23 model was co-funded by ETH Zurich (contract ID 19203), the Federal Office for Civil Protection (FOCP; contract no. 353007634), and the Federal Office for the Environment (FOEN; contract no. 16.0115.PJ /
Q114-1464). The development of communication products also benefitted from partial funding from European Union's Horizon 2020 research and innovation program under grant agreement Real-time earthquake rIsk reduction for a reSilient Europe (RISE) project (http://www.rise-eu.org) No. 821115.

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
