# Peer review of "The Earthquake Risk Model of Switzerland ERM-CH23"

_EGUsphere, 2023_

## Author Comment (AC1)

**REVIEWER 1**

The paper discusses the key issues of the Earthquake Risk Model of Switzerland (ERM-CH23). It is the result of huge work from the hazard, vulnerability and exposure sides. Authors summarize the main input and output of the study, including the comparison with other models. Such large research effort typically goes in research reports because a journal paper is often too short and also these projects implement state of the art research without actual scientific innovation. This is also the case of this manuscript in my opinion. Reproducibility is also impossible based on this work. Therefore I regret that I cannot recommend this manuscript for publication until it qualifies as a research paper. This notwithstanding, I appreciate the work and understand many of the authors are directly involved in the special issue this manuscript is submitted to. I also offer some comments as listed.

We appreciate the reviewer's perspective and acknowledgement of the significant effort involved in the development of the Earthquake Risk Model of Switzerland (ERM-CH23). Describing the entirety of the model's development within a single article is indeed challenging. As the reviewer suggests, a comprehensive research report has been published and is referenced throughout the manuscript, offering readers a detailed resource for further clarification.

However, we respectfully disagree with the notion that articles like this lack value in the scientific literature, for a number of reasons:

a) While this article only summarizes the key points of the source, ground motion and vulnerability components that underpin ERM-CH23, it provides a rather extensive overview and discussion of its key results. The quantification of earthquake risk in Switzerland, at this level of detail, is by itself a novel piece of information that was previously lacking from the public domain. ERM-CH23 combines rich datasets and the expertise of different actors to provide the most up-to-date view on earthquake risk in Switzerland. The latter includes the level, frequency and spatial distribution of different loss types. Moreover, we argue that this fits perfectly with the scope of this special issue: *"We invite contributions to a virtual special issue in NHESS related to the European seismic hazard and risk model building, model results, sensitivity analysis, and all model components. We also welcome contributions on seismic hazard and risk assessment in general, at the local, national, and regional levels"*

b) Besides the results and products, the manuscript also provides a summary of ERM-CH23, highlighting its key features. Summary articles (in the context of complex seismic hazard/risk models) make it significantly easier for potentially interested readers to acquire basic knowledge and understanding of the model. If needed, readers can subsequently search for specific (and relevant to them) clarifications in the ~200 page report, rather than blindly going through it.

Similar articles are relatively common practice for such large PSHA or regional risk assessment efforts and are well-received by the scientific community as evidenced by the number of citations they receive. See for example:

- Field, E.H., Jordan, T.H., Page MT, et al. (2017). A Synoptic View of the Third Uniform California Earthquake Rupture Forecast (UCERF3). Seismological Research Letters 88(5): 1259–1267.

- Silva, V., Amo-Oduro, D., Calderon, A., et al. (2020). Development of a global seismic risk model. Earthquake Spectra 36: 372–394.

- Dolce, M., Prota, A., Borzi, B. et al. (2021.) Seismic risk assessment of residential buildings in Italy. Bull Earthquake Eng 19, 2999–3032. https://doi.org/10.1007/s10518-020-01009-5

c)  ERM-CH23 is a national model intended to facilitate risk management and civil protection in Switzerland. Besides this being the focus of this very special issue, there is increased wider interest in the development of such models in collaboration with national authorities and their use for risk mitigation. Therefore, ensuring the visibility of ERM-CH23 is important. Like similar national/regional models, it will also serve as a reference or benchmark for the scientific community (both in terms of its obtained results for Switzerland as a piece of the global seismic landscape, but also methodologically).

Moreover, catastrophe risk models are more than the sum of their components and their development requires significant inter-disciplinary expertise. At the very least in the experience of the first author, articles like this provide valuable context for young researchers by helping them 1) establish reference points beyond their individual and very specific area of research, 2) acquaint themselves with assumptions and practices typically followed in such contexts, 3) understand the interdependencies between disciplines to develop societally relevant models and assets

1. No equations are reported in the paper. I understand that the paper discusses the results of a huge work involving several expertise and alternative approaches (combined by the logic tree) but an analytical framework should be described.

   This paper focuses on the earthquake risk assessment for the Swiss territory. The earthquake risk assessment framework that is adopted here is rather standard and discussed in multiple (maybe even hundreds nowadays) studies that have used the OpenQuake engine. It is also very unclear from this comment what specific equations are missing and would improve the manuscript should they have been in there.

2. The introduction section should briefly present the structure of the paper.

   We will amend the last sentence of the introduction to the following:

*"In the following sections, we give an overview of ERM-CH23 and its subcomponents (seismic source, ground motion, site amplification, fragility, consequence and exposure models) and then present primary results and insights such as comparison with other published models."*

3. Figure 1 is never cited in the body of the text.

   Thank you for noticing, we will cite it.

4. It is discussed that ERM-CH23 relies on 2015 Seismic Hazard Model (SUIhaz2015; Wiemer et al., 2016), and that some adjustments were made to it for use within the context of ERM-CH23 (lines 65-69). In turn, SUIhaz2015 relies on ESHM13, among others. This reviewer wonders whether one additional adjustment to SUIhaz2015 may be the replacing of ESHM 13 with ESHM20 (Danciu et al., 2021). Can authors comment on it?

   As is the case with any model, improvements and adjustments will always be needed with time, as science progresses and more data becomes available. While this suggestion is certainly a reasonable one, it was not within the scope of ERM-CH23 to adopt a different source model than the one used as part of the authoritative national seismic hazard model (SUIhaz15).

   The adjustments made here were minor (use of mean source model for reasons explained in the manuscript, minimum magnitude set at 4.5 instead of 4.0, see response to later comment for justification) and aimed purely at tailoring the source model for application in risk assessment, rather than adopting new science. Moreover, compatibility between the national hazard and the national risk model was another reason that led to the decision, at the beginning of the project, in 2017, not to replace the components of SUIhaz2015.

5. Authors also discuss that ERM-CH23 is based on two main "sub-models". One uses spectral accelerations, the other adopts macroseisimic intensity, and it seems that they are arranged in a logic tree framework. However, it is not clear, at least to this reviewer, how hazard results (hazard curves), in terms of different ground motion intensity measures, are finally combined.

   ERM-CH23 only provides risk estimates. Hazard curves were not computed since (1) they were not an intended product of ERM-CH23, and (2) they are not required within the standard event-based risk assessment workflow that is almost universally adopted for such (regional risk assessment) applications.

   For a better understanding of the latter, we invite the the reviewer to consult the documentation of the OpenQuake engine:

- Pagani, M., Monelli, D., Weatherill, G. A. and Garcia, J. (2014). The OpenQuake-engine Book: Hazard. Global Earthquake Model (GEM) Technical Report 2014-08, doi: 10.13117/- GEM.OPENQUAKE.TR2014.08, 67 pages

- Crowley, H., and Silva, V. (2013). OpenQuake Engine Book: Risk v1.0.0. GEM Foundation, Pavia, Italy.

- https://docs.openquake.org/oq-engine/manual/latest/

Some other relevant references are also listed below:

- Crowley, H., and Bommer, J.J. (2006). Modelling seismic hazard in earthquake loss models with spatially distributed exposure. Bulletin of Earthquake Engineering 4(3): 249–273.

- Silva, V. (2018). Critical Issues on Probabilistic Earthquake Loss Assessment. Journal of Earthquake Engineering, Taylor & Francis 22(9): 1683–1709.

- Baker, J. W., Bradley, B., and Stafford, P.. (2021). Seismic Hazard and Risk Analysis. Cambridge, United Kingdom: Cambridge University Press.

- Mitchell-Wallace, K., Jones, M., Hillier, J., et al. (2017). Natural catastrophe risk management and modelling. 1st ed. Natural catastrophe risk management and modelling: A practitioner's Guide, Hoboken, NJ: John Wiley & Sons, Ltd.

6. What is the considered minimum magnitude? Lines 81-83, events of magnitude around 4 have been known to cause damage and casualties elsewhere, the justification for minimum magnitude 4.5 needs to be contextualized for Switzerland or justified quantitatively.

   As stated on line 83, the minimum considered magnitude was 4.5.

7. Line 84. But they are the most frequent earthquakes. Since the cited paper refers to PSHA, can the authors demonstrate that the neglected magnitudes do not contribute in terms of losses?

   a. As the reviewer points out in the previous comment, small earthquakes can occasionally cause damage, especially if they occur at shallow depths and in proximity of exposure concentrations. For this reason, ERM-CH23 adopts a relatively small minimum magnitude (Mw 4.5). For comparison, we overlay the assumptions of other recent seismic risk studies:

| Reference | Minimum considered | Region |
|---|---|---|

| | magnitude | |
|---|---|---|
| ESRM20; (Crowley et al., 2021) | 5.00 | Europe |
| GEM 2018; (Silva et al., 2020) | 5.50 | Global |
| Canadian Seismic Risk Model (CanSRM1) (Hobbs et al., 2023) | 4.5 – 5.0 (depending on tectonic regime) | Canada |
| (Kohrangi et al., 2021) | 4.5 | Iran |
| (Rao et al., 2017) | 5.0 | California |
| (Salgado-Gálvez et al., 2023) | 5.0 | Caribbean and Central America |
| (Goda et al., 2016) | 4.5 | Malawi |

- Crowley H, Dabbeek J, Despotaki V, et al. (2021) European Seismic Risk Model (ESRM20).
- Goda K, Gibson ED, Smith HR, et al. (2016) Seismic risk assessment of urban and rural settlements around lake malawi. Frontiers in Built Environment 2(November).
- Hobbs TE, Journeay JM, Rao AS, et al. (2023) A national seismic risk model for Canada: Methodology and scientific basis. Earthquake Spectra: 875529302311734.
- Kohrangi M, Bazzurro P and Vamvatsikos D (2021) Seismic risk and loss estimation for the building stock in Isfahan : part II — hazard analysis and risk assessment. Bulletin of Earthquake Engineering, Springer Netherlands (0123456789).
- Rao AS, Weatherill GA, Silva V, et al. (2017) Beyond Button Pushing — Seismic Risk Assessment for California. Pavia, Italy.
- Salgado-Gálvez MA, Ordaz M, Singh SK, et al. (2023) A Caribbean and Central America Seismic Hazard Model for Sovereign Parametric Insurance Coverage. Bulletin of the Seismological Society of America 113(1): 1–22.
- Silva V, Amo-Oduro D, Calderon A, et al. (2020) Development of a global seismic risk model. Earthquake Spectra 36(1_suppl): 372–394.

b. An analysis of small damaging earthquakes is provided by Nievas et al. (2020). Therein (Fig 13), the authors estimate the proportion of damaging M4.00-M4.25 earthquakes at ~0.2% and of damaging M4.25-M4.50 earthquakes at ~1%. Now these are global statistics and arguably subject to many uncertainties and likely ascertainment bias. But let's take them as a ballpark estimate for the sake of this

argument, and also assume a Gutenberg-Richter magnitude frequency distribution with a b-value of 1.0:

With the b=1 assumption, for every 200 earthquakes with M>4.25, 112 of them will, on average, be in the 4.25-4.50 range. If we assume that 1% of the M4.25-M4.50 earthquakes will be damaging, that means that we will have about 1 of those small earthquakes cause some damage. This damage (as intuitively expected, and also informed by sensitivity checks carried out as part of ERM-CH23) is going to be minimal and several orders of magnitude smaller than the damage caused by M>6. For reference, under the same b=1 assumption, around ~3-4 of the 200 earthquakes would be expected to be >M6, and all of them would cause significant loss.

Therefore, the contribution of such small and rarely damaging earthquakes is expected to be minimal (see also following point).

Nievas CI, Bommer JJ, Crowley H, et al. (2020) Global occurrence and impact of small-to-medium magnitude earthquakes: a statistical analysis. Bulletin of Earthquake Engineering, Springer Netherlands.

c. The figure below (also available in the ERM-CH23 report; Wiemer et al., 2023), shows the contribution of different magnitude earthquakes to the total structural/nonstructural economic average annual loss (AAL). We see that the lower magnitude bin (4.5 – 5.0) in this chart provides a contribution of around 5%. Earthquakes of magnitude <4.5, if included, would contribute significantly less than that towards the total AAL.

[Figure]

d. Lastly, there are also practical considerations when setting up a risk model, in particular at a national level. As it can be understood from the previously laid out statistics, including such small earthquakes would greatly increase the computational cost of the analysis. In sensitivity checks carried out during model development with a subset of the model, setting $M_{min}$ to 4.7 led to 4-5 times shorter runtime compared to setting $M_{min}$ to 4.0. It is unclear what the difference would be for the whole model run (and at the moment we do not have the exact time gain for $M_{min}$=4.5), but it is clear that this is an important consideration, especially when (1) the effect of such small earthquakes is minimal (see previous point) and (2) the model runtime with $M_{min}$ =4.5 is already in the order of a bit less than a week on a 256-core machine.

8. In the discussion of section 3.1.1, an important piece of information seems to be missing (or if it is implied, then it is unclear) and that is the weights assigned to the five branches. In fact, the first and last quantiles seem extreme enough to warrant justification if equal weights were adopted.

No, the weights were not equal. The sentence will be amended to the following:

"*From the ensemble earthquake rate model, the activity rates corresponding to the 2.5%, 16%, 50%, 84%, 97.5% quantiles were obtained and assigned as five alternative logic tree branches with weights corresponding to the associated area under a normal distribution that they represent. For further details on the source model of SUIhaz2015, the reader is referred to Wiemer et al. (2016).*"

9. Section GG. Was spatial correlation motion of GMM residuals considered in the analyses?

Spatial correlation of GMM intra-event residuals was not considered. The inter-event residual was sampled once per event (therefore accounting for this source of GM correlation).

The spatial correlation of intra-event ground motion residuals is important for the risk assessment of spatially distributed portfolios. Several studies have shown that neglecting such spatial correlation tends to overestimate frequent losses and underestimate rare ones (Markhvida et al., 2017; Park et al., 2007; Weatherill et al., 2015). It is even more important when assessing interconnected systems such as transportation networks (Costa et al., 2018).

That said, it is usually not accounted for in large scale risk calculations (Crowley et al., 2021; Silva et al., 2020) for a number of reasons. Firstly, it is computationally difficult

to invert a covariance matrix of tens or hundreds of thousands points and also very prone to numerical instabilities. Therefore, including spatial correlation for such large size calculation is not possible with the OpenQuake engine. Moreover, in the case of risk analyses with aggregated exposure (rather than individual assets), modelling spatial correlation is a controversial practice and, in some cases, it has been shown to overestimate the true correlation (Bazzurro and Park, 2007; Stafford, 2012). Our own sensitivity analysis supported this notion.

- Bazzurro P and Park J (2007) The effects of portfolio manipulation on earthquake portfolio loss estimates. In: 10th international conference on applications of statistics and probability in civil engineering, Tokyo, Japan.
- Costa C, Silva V and Bazzurro P (2018) Assessing the impact of earthquake scenarios in transportation networks: the Portuguese mining factory case study. Bulletin of Earthquake Engineering, Springer Netherlands 16(3): 1137–1163.
- Crowley H, Dabbeek J, Despotaki V, et al. (2021) European Seismic Risk Model (ESRM20).
- Markhvida M, Ceferino L and Baker J (2017) Effect of ground motion correlation on regional seismic loss estimation : application to Lima , Peru using a cross-correlated principal component analysis model. 12th International Conference on Structural Safety and Reliability (June): 1844–1853.
- Park J, Bazzurro P and Baker JW (2007) Modeling spatial correlation of ground motion Intensity Measures for regional seismic hazard and portfolio loss estimation. In: Applications of Statistics and Probability in Civil Engineering.
- Silva V, Amo-Oduro D, Calderon A, et al. (2020) Development of a global seismic risk model. Earthquake Spectra (February): 875529301989995.
- Stafford PJ (2012) Evaluation of structural performance in the immediate aftermath of an earthquake: a case study of the 2011 Christchurch earthquake. International Journal of Forensic Engineering 1(1): 58.
- Weatherill G a., Silva V, Crowley H, et al. (2015) Exploring the impact of spatial correlations and uncertainties for portfolio analysis in probabilistic seismic loss estimation. Bulletin of Earthquake Engineering: 957–981.

10. Line 79. when later is used as an object to the sentence, the "t" is doubled.

    It will be corrected, thank you.

11. Line 90. the reference Zhao et al. (2006) must be replaced by Zhao (2006).

    It will be corrected, thank you.

12. Line 93. Wiemer is cited twice.

    It will be corrected, thank you.

13. Table 1 lists, among others, two ECOS09 models, but they are not even mentioned in the paper (the not expert readers do not know that ECO09 is the Earthquake Catalogue of Switzerland). The other two models are calibrated based on data from metropolitan France (Baumont et al., 2018) and central Asia (Bindi et al., 2011). Can authors motivate such a choice?

Table 1 contains the reference for the ECOS09 models right next to their abbreviation and weight.

The IPEs were selected primarily based on the results of a residual analysis conducted on the macroseismic dataset for the region (see Wiemer et al. 2023). Among the top performing IPEs in terms of loglikelihood scores, we then made a selection accounting also for other criteria, such as having a set that encompasses different views (e.g. calibrated on Swiss data vs non-Swiss data, different attenuation behavior, different amplitude at short-distances).

The text before the table will be modified to the following:

*"A residual analysis was conducted on the macroseismic dataset for the region, in order to compare a collection of candidate IPEs. The latter were ranked and four of the best performing ones were then selected (Table 1) to represent the body, center and range of intensity data. Besides the residual analysis, considerations were made to compile a set that encompasses different epistemic views (e.g. amplitude and attenuation trends) with members calibrated on different datasets (the two ECOS-09 models are fit on Swiss macroseismic observations, whereas the other two on wider datasets)."*

14. The definitions of PGV and SA do not mention how the two horizontal components of motion are combined.

The fact that the geometric mean is used will be added in the site amplification section where the intensity measures are first introduced.

15. In figure 3, vulnerability curves expressed in terms of two different intensity measures (spectral acceleration at two different vibration periods) are presented in a single panel. This should be avoided, as it is typical for the reader to compare vulnerability by the relative shift of the curves along the abscissa (as the authors themselves do in section 3.2.4) and thus, grouping curves this way can be misleading. Also: in the caption spectral acceleration is denoted both as SA and Sa.

Sure, but we don't think this is a big issue here because (1) the SA(0.6) curves are a minority and refer to less frequent typologies that are not even annotated in the figure

(gray lines under label 'other') and (2) we explicitly make the distinction through the linestyle (dashed vs continuous) and associated legend and axis label.

Thank you for noticing the caption, it will be corrected.

16. Between fragility functions and consequence models (3.2.2 and 3.2.3) some mention of damage states seems to be missing?

Thank you for the suggestion. We will amend the last sentence of the fragility function section to indicate that we are using the EMS-98 damage scale. The damage scale is again mentioned in the consequence model section.

17. Line 240. The fact that earthquake recurrence is modelled via a Poisson stochastic process has nothing to do with whether or not "aleatory uncertainties" are considered in its modelling.

Nowhere in the manuscript is such a thing stated. All that is stated is that earthquake recurrence (aleatory uncertainty) is modelled via a Poisson process. In any case we will amend that sentence to the following:

"*As usual, aleatory uncertainties are considered in the modelling of earthquake occurrence (modelled as a Poisson process), in the modelling of ground motion (modelled with a lognormal distribution for ground motion or a normal distribution for intensity), as well as in the modelling of loss given ground motion (modelled with a beta distribution).*"

18. Line 244. This makes results not replicable. In other similar situations, an "average" branch able to approximate the results of the whole logic three was identified/constructed. Did the authors consider such opportunity? Moreover, it may be interesting to deepen the discussion of results describing the logic three branches that produce the lowest and the largest results.

It is not possible to run a full enumeration of all 165,888 branches of the model. The 424 branches we considered took almost a week on our 256-core CPU / 2TB RAM machine. Similar strategies have been used in other models (SUIhaz15, ESHM20, ESRM20, etc).

The strategy of selecting a branch which returns results that are the closest to the mean is undoubtedly a very pragmatic one, and appropriate in many occasions. However, it is not ideal in many other. For instance, a branch might be selected on the grounds of best approximating the mean country-wide loss exceedance probability curve. The same branch however might not be always close to the mean at local scales (e.g. canton or municipality). Likewise it might not be close to the mean for

some of the loss types considered (structural/nonstructural, contents, fatalities, injuries, displaced population). The selection might be carried out with multiple targets in mind, but it is often difficult to find a branch to satisfy all of them (plus unknown targets, e.g. if one was to run the model in the future with a specific building portfolio).

19. Lines 250-251. Monte-Carlo simulation cannot be said to be "required" in this context. It is simply a matter of choice, and the authors would be well-advised to have the courtesy of explaining the motivation behind said choice, while being more careful in their phrasing.

The use of Monte Carlo simulations for this purpose has long been established as the golden standard in such applications. However, the reviewer is right in the sense that what we primarily wanted to say here is that "an event-based approach" (i.e. a framework that starts with a set/catalogue of earthquake ruptures rather than an analytical integration like in PSHA) is required[1].

From the event-set forward, Monte Carlo simulation is typically used for calculation of ground motions and associated loss. However, there are indeed a couple of approaches that do not necessitate it, the most notable being the ones proposed by Ordaz et al. (2000) and Wesson et al. (2009). The approach of Ordaz et al. (2000) involves certain rough assumptions for what concerns the correlation of losses across the portfolio, whereas the Wesson et al. (2009) method has not at the moment propagated to a software implementation (at least in the public domain) that we could use.

- Ordaz M., Miranda E., Reinoso E., Perez-Rocha L.E. (2000), Seismic Loss Estimation Model for Mexico City, 12th World Conference on Earthquake Engineering.
- Wesson RL, Perkins DM, Luco N, et al. (2009) Direct calculation of the probability distribution for earthquake losses to a portfolio. Earthquake Spectra 25(3): 687–706.
* * *
[1] This is not particularly ambiguous; integration-based approaches cannot properly capture the correlation of loss at different sites of interest. It has been discussed several times in the literature, two (of the many possible) excerpts are laid out below:

Baker et al., 2021 (pg. 469, section 11.4): *"Because there are often many sites of interest, it is __not feasible__ to evaluate the high-dimensional joint distribution of IM over all values in the integral of Equation 11.2. Instead, we usually __use Monte Carlo simulation__ to sample rupture scenarios and then sampled from the joint IM distribution, conditional upon each rupture."*

Crowley and Bommer, 2006 (pg 270-271): *"The aim of this paper has been to compare the results of a loss model using two different procedures to represent the seismic hazard: the use of conventional PSHA to obtain hazard curves at many locations, and the use of multiple earthquake scenarios, defined through **MCS** (Monte Carlo Simulation) based on the seismicity model, to generate the ground motions at all sites of interest. The two procedures have been seen to produce **very different loss exceedance curves for a geographically distributed building stock**, even though the same seismic hazard is represented in both methods. […] it is clear that **the use of PSHA should nonetheless be viewed as a compromise**."*

Some other approaches also exist and are used by CAT model vendors, but besides the fact that those are generally not public, they typically involve approximations that aim to reduce computational cost, while staying as close as possible to the results one would obtain from a full-blown Monte Carlo simulation. But even in that space, there is a trend to move towards simulation-based frameworks as exemplified by the push towards solutions such as the one implemented in the OASIS loss modelling framework (https://oasislmf.org/).

Following the above, we will rephrase that sentence to:

"To assess the earthquake risk over a spatially distributed exposure, a so-called event-based approach is required that starts with the generation of stochastic earthquake catalogues. Usually, this is followed by the simulation of associated random ground motion fields for each generated earthquake rupture"

20. In the opinion of this reviewer, section 3.4 should be moved after section 3.1.3, as it deals with the logic tree at the basis of the hazard assessment.

The logic tree also contains branches related to the exposure modelling, so our preference is to keep it after the relevant section.

21. With reference to Figure 6 (showing the logic tree): what is the building mapping scheme? And what do RB and RF mean?

The building mapping scheme refers to the process of associating a certain structural typology to each building in the database. The two branches RB and RF have been defined beforehand in lines 214-215.

22. Figure 6 the SUIhaz15 logic tree branching is illegible and beyond the pdf file resolution.

The SUIhaz15 logic tree can be found in the relevant report as cited in the manuscript. We will also added as an appendix to this article. It is simply impossible to provide a legible 16 x 16 x 18 x 18 GMPE tree in a single figure. The icon in Figure 6 is meant as a symbol and not for reading through.

23. Line 256. "probable maximum loss" lacks a definition (while it emphatically needs one).

We generally agree that the term is often misused and/or used to refer to different things. Since at least in the scientific literature it is often used to refer to loss vs return period plots, we keep this definition and add it in parenthesis as follows:

*"probable maximum loss (PML; herein defined as loss versus return period of exceedance)"*

24. Line 262. The term "epistemic distribution" is a neologism that should be avoided.

English is not our first language and therefore we are open to correcting any linguistic mishaps. That said, it is not obvious to us what the problem with this term is and what alternative the reviewer prefers. In our view, it portrays accurately what we want to convey. We would kindly ask the reviewer to be a bit more precise if he still thinks there is some problem with this term.

Moreover, a quick search in the literature yielded several publications where this term is used. For instance, below we list a couple of examples of publications where this term is used, one by senior authors from the seismological community and one by senior authors from the engineering community:

● Gerstenberger, M. C., Marzocchi, W., Allen, T., Pagani, M., Adams, J., Danciu, L., et al. (2020) Probabilistic Seismic Hazard Analysis at Regional and National Scales: State of the Art and Future Challenges. Reviews of Geophysics 58(2): 0–3.
● Liel AB, Haselton CB, Deierlein GG, Baker JW (2009) Incorporating modeling uncertainties in the assessment of seismic collapse risk of buildings. Structural Safety, Elsevier Ltd 31(2): 197–211.

25. According to reviewer some additional information should be provided about bilinear capacity curve used in fragility assessment leaving to related papers the description of the numerical models.

We appreciate the reviewer's point and thought about adding further information on that too. However, due to the different sources and methodological aspects (e.g. for different building materials), such a section would end up very long and tedious for the reader. Ideally it should have been published as a separate paper, however it is exhaustively described in the technical report of the model. Hence we will add a reference to that in the text: "*Further details on the definition of capacity curves and the methodology followed to derive fragility curves can be found in (Wiemer et al., 2023).*"

26. Method used to derive fragility functions should be cited at least. Leaving it to the related paper makes the manuscript difficult to read.

It is indeed cited (lines 140-141, line 145).

27. How modeling uncertainties in fragility assessment have been accounted for in risk assessment? Paragraph 3.4 does not seem to treat this issue.

It is not very clear what is meant here as "modeling uncertainties". Typically this term is used to refer to uncertainties regarding the approach used to modeling the response of a structure (e.g. finite-element modeling approach) and sometimes to uncertainties pertaining to the modeling of specific subcomponents (e.g. hysteresis models). Here, the development of the fragility model is based on archetype capacity curves and an R-μ-T relationship as proposed in Michel et al. (2018). Uncertainties such as material uncertainty and building-to-bulding variability are considered through the stochastic generation of multiple capacity curves characterizing single degree of freedom systems.

- Michel C., Crowley H., Hannewald P., Lestuzzi P., Fäh D. (2018). Deriving fragility functions from bilinearized capacity curves for earthquake scenario modelling using the conditional spectrum, Bulletin of Earthquake Engineering

28. One would expect some more in depth discussion about the differences seen in Figure 7 between the SAM and MIM model risk estimates.

We will add a few additional sentences in the discussion of Figures 7-9. The paragraph below Figure 8 will now read:

"*Of course, as shown in both Figure 7 and Figure 8, there is non-negligible dispersion around the mean estimates reported above, which reflects the large uncertainties in many parts of the model. The main driver of the epistemic uncertainty is the modelling of ground shaking as indicated by the tornado diagrams (Porter et al., 2002) in Figure 9, an observation that is in line with previous studies (e.g. Field et al., 2020). For structural/non-structural AAL, the choice of IPE/GMM leads to a ~5-fold difference, whereas for fatality AAL the difference is ~4-fold for IPEs and 35-fold for GMMs. Important differences are also observed between the two submodels, MIM and SAM, especially for fatalities, as also seen in Figure 7. These large differences in estimated fatalities are attributed to a combination of factors. Fatalities are primarily driven by structural collapses, therefore differences at the least well constrained parts of ground motion and fragility models (i.e. ground motion and intensity amplitudes at short source-to-site distances, collapse fragility functions) manifest in divergent estimates, which reflect the large uncertainty in the estimation of human losses. Lastly, the building mapping scheme and site amplification uncertainties explain a smaller part of the total uncertainty around the country-wide AAL. That said, note that even the latter two sources of epistemic uncertainty might lead to significant differences at local scales (see Wiemer et al., 2023), making their inclusion in the model very important.*"

29. Caption to figure 9 needs some revising, should that be "epistemic variability"? The meaning of the sentence "if only…" is not clear. More in general, the figure is not clearly discussed in the text of the paper.

The caption will be amended to the following:

*"Figure 9. Epistemic uncertainty tornado diagrams for structural/non-structural (top) and fatality (bottom) AAL. The bars for each branching level show the minimum and maximum AAL estimate obtained by varying only the input at that level (e.g. only the amplification branch) while keeping the rest of the logic tree unchanged.. The MIM and SAM specific bars refer to estimates of those sub-models rather than of the entire model. Finally, in the case of GMMs, since enumeration is not possible and 400 branches are sampled, the bars simply refer to the minimum and maximum values obtained across these 400 samples."*

Hopefully this improves its clarity. The figure is fully discussed in the paragraph preceding it.

30. Line 359. some additional details about the tools should be provided.

It is not clear what the reviewer means by tools. The risk assessment was carried out with the OpenQuake engine, as detailed in section 4.

---

## Author Comment (AC2)

**REVIEWER 2**

This manuscript perfectly describes the aggregation of all current knowledge and methods for characterizing seismic hazard, vulnerability and risk, applied to Switzerland. This manuscript compiles several decades of work, from different groups (seismology, engineering seismology, earthquake engineering and even communications), for a single product as target: the seismic risk map of Switzerland. The authors are all experts in their field, and have contributed in one way or another to the definition of one of the strategies in their area of expertise followed in this study: definition of source zones, selection of GMMs and associated logic tree, characterization of site effects, definition of vulnerability curves and loss models. Each step references a series of peer-reviewed publications, giving scientific validity to the study.

So why this article?

1. On reading this manuscript, I fail to see the scientific contribution or the added value for publication in NHESS. Throughout the manuscript, reference is made at every step to Wiemer et al. 2023, and the question arises as to how this article compares with Wiemer's paper and what is the original contribution with respect to this 2023 paper?.

   We would like to primarily refer to our reply to the first reviewer's general assessment. But in short and without repeating our other response, we see the following as contributions of this article:

   · Quantification of earthquake risk estimates for Switzerland for different loss types (economic building and contents loss, fatalities, injuries, displaced population); breakdown by canton or municipality (Fig 10); breakdown by building typology (Fig 11).

   · Presentation of the multidisciplinary approach adopted by the Swiss Seismological Service for the development of Switzerland's national earthquake risk model. Summary of key elements and references to the technical report for interested readers.

   · Various insights, e.g. discussion and statistics of the composition of the modelled building exposure, differences with other models and reasons for them (we will add a couple of additional plots, see response to comment 2.), communication concept.

Wiemer et al. (2023) is the ~200 page technical report that describes the model in detail. We refer to it in the same way we would to an electronic supplement, since the full details of a model of this scale cannot be summarized in a short article.

2. A few attempts at "testing" with recently published models show that, in the end, the estimate is neither right nor wrong, good nor false,… but rather overestimates one and underestimates the other, without it being clear why. This testing section could have been extended, which would have improved the novelty of this manuscript: by identifying the source of the differences, by discussing the need for a regional model versus a local model, by discussing some options taken (notably in ESRM20) or by extending the comparison between models to specific target areas for which the site conditions and the exposure model are (perfectly!) known (e.g. Valais or maybe the Basel urban area).

We will extend the comparison section by adding the figure below to reinforce the points made in the pre-existing paragraph (difference in exposure values, resolution of site amplification model).

[Figure]

Figure 13. Comparison of total structural replacement cost between ESRM20 and ERM-CH23 (left); Ratio of 475-year SA (0.3 s) values on soil predicted by the SAM component of ERM-CH23 and ESRM20 across Switzerland.

With respect to the reviewer's last point, unfortunately, there are no specific areas where either the site conditions or the exposure model are perfectly known. The site model might be more reliable in certain areas, although still uncertain and ground motion will still be largely impacted by the underlying GMMs. The exposure model would also be much more uncertain if we were to focus on a smaller region, since building materials are assigned based on survey statistics. For larger exposures, the estimates are likely to be robust due to the law of large numbers and averaging out of errors. But for smaller areas, they would be less reliable, not necessarily more.

3. The issue on the contribution of each step in the final uncertainties could also have been developed. We've known for some time that the uncertainty linked to the hazard assessment is certainly the factor with the greatest impact: this message now seems self-evident, but has yet to be confirmed. In particular, this may be due to a lack of understanding of vulnerability and exposure: if we were to benefit from the same amount of experimental data as those used for ground motion modelling (the number of which has increased enormously over the last 20 years), wouldn't we end up with a shared contribution of hazard and exposure to the final uncertainty? Or is the variability of ground motion intrinsically (non-epistemic) greater than that of structural response? This question, debated and simulated thanks to the logic trees defined herein, could have been tested, bringing a novelty to this study.

We agree with the reviewer that vulnerability (and to, likely, some lesser degree exposure) modeling involves a great deal of uncertainty that is currently not being modelled. We do not necessarily agree with the notion that hazard uncertainty should be expected to completely overpower any uncertainty that would be expected from vulnerability modeling, were the latter modelled thoroughly. Some of the challenges and size of uncertainties in vulnerability assessment are for instance discussed in (Cremen and Baker, 2020; Silva et al., 2019).

- Cremen G and Baker JW (2020) Variance-based sensitivity analyses and uncertainty quantification for FEMA P-58 consequence predictions. Earthquake Engineering and Structural Dynamics (May 2019): 1–20.
- Silva V, Akkar S, Baker J, et al. (2019) Current challenges and future trends in analytical fragility and vulnerability modeling. Earthquake Spectra 35(4): 1927–1952.

Modelling the epistemic uncertainty on the vulnerability component is still a barrier that the earthquake risk modelling community (and us here) has not overcome (for large scale risk studies). Before delving into that though, we would like to stress that we are referring to systematic differences (bias) to the mean vulnerability function that describes the behavior of each building class. The reason for this bias could be methodological/modeling assumptions, unrepresentative samples or building archetypes, etc. Other types of uncertainties that characterize individual buildings (e.g. material uncertainty) but are not systematic (building-class wide), might have a huge effect on the risk estimates of individual buildings (or small building portfolios), however they are expected to have small influence on the regional results (since these kind of errors average out, as long as the mean vulnerability curve is unbiased).

To model the first type of systematic uncertainties (e.g. as a logic tree) there are the following obstacles:

- The development of vulnerability models is typically analytical, since empirical data (even more so for Switzerland) are lacking. Therefore, it requires a huge amount of effort and domains of expertise (concrete, steel, masonry, timber, etc buildings) that usually lie with different groups.
- One could envisage introducing some uncertainty based on engineering judgment by shifting the original vulnerability curves to the left or right with appropriate weights. This can be done if one thinks there could be systematic biases in the methodology used to derive the fragility model (although it is difficult to guess the direction and magnitude of such possible bias). On the other hand, (arguably much larger) uncertainties that are not tied to the methodology but to the data and/or assumptions used for modeling different building types would be independent. Therefore, one would need to create multiple logic tree branches where let's say an upper vulnerability model for reinforced concrete mid-rise is combined with a lower vulnerability model for stone masonry low-rise, etc. These combinations are intractable and difficult to model. Sampling could be an option, although (1) OpenQuake at the moment does not allow for that, (2) convergence issues would then arise when looking at the results by building typology.
- In any case, none of this would address directly the reviewer's comment, since it would be based purely on judgement and assumptions rather than on data, which, as the reviewer also says, are missing when it comes to vulnerability modeling.

Finally, we should say that regardless of the above, testing any of that would be a paper of its own. This is outside our scope here, which is the presentation of the ERM-CH23 model and mainly of the obtained estimates of seismic risk in Switzerland.

4. Finally, by compiling and aggregating previously published studies, much information is missing in order to follow the flow of the manuscript: GMM equations are missing, parameters characterizing site conditions are missing, descriptions of vulnerability models are missing etc... and in the end, the paper looks like a compilation of studies, not a scientific paper.

- We will an Appendix with the GMM equations comprising the logic tree.
- The parameters characterizing the site conditions are actually mentioned in the paper.
- We will greatly expand the section on the developed consequence model (see response to comment 10.).

All in all, we fully understand the reviewer's perspective in this comment, although it is the nature of this work (risk modeling in general) that relies on multiple components that come from other studies and cannot be summarized in a single article.

For instance, as requested, we have added an appendix with the GMM equations used. These have been published in scientific articles and are still not summarized here (and it would not be possible to summarize 18 GMMs and 4 IPEs). Likewise, almost every published study on seismic hazard and risk assessment cites the used GMM equations without describing them further.

The site amplification model developed for ERM-CH23 has also recently been published in a peer-reviewed journal. Instead of just citing like we do with the GMMs, a high-level overview is provided (because of its direct relevance; and because of the fact it is a single model rather than 18, and hence easier to give an overview). Perhaps this gives the impression of missing information, but is it really much different?

That said, again, we want to state that we understand the perspective of the reviewer, but if model components cannot be summarized and referred to other publications, it would be impossible to publish articles on seismic risk. And while methodologically, this particular paper, constitutes primarily an application of the state-of-the-art risk framework, it combines the various individual high-quality components (e.g. amplification layer, exposure) derived as part of the ERM-CH23 project (some of which derived through novel approaches), to reach an updated view of seismic risk for the country of Switzerland (e.g. compared to ESRM20 or GEM that at the time used less detailed datasets for Switzerland). Lastly, we would argue that our paper (together with the cited report) gives significantly more information than other recent similar publications.

Some general comments:

5. Introduction lines 22-27: very classic, and in a special issue like this one, we're convincing the convinced. In my opinion, a little added value, such as the fact that natural hazard management programs save $3-4 on average for every $1 invested, goes a long way towards justifying the production of seismic risk maps like the one presented here.

We thank the reviewer for his suggestion. We will modify the opening paragraph to add the reference to the study of NIBS that we were not previously aware of:

"*Natural hazards can cause widespread damage, loss of life, and disruption to critical services such as water, power, and transportation. Earthquake risk mitigation programs are effective, and as cities and populations continue to expand, they become increasingly vital to safeguard lives, infrastructure, and economic stability. A study from the National Institute of Building Sciences in the US estimated that Federal Mitigation Grants for earthquake hazards save 3 dollars per 1 dollar spent (National Institute of Building Sciences, 2019). Catastrophe risk models, in particular, can aid governments and other stakeholders in [...]*"

6. Line 52-55: " ... is considered moderate with three to four earthquakes a day.... " This sentence (and many others) is too vague, certainly because of the systematic reference to already published papers. Relying on previous studies should not compromise understanding of the article as a whole.

While we did not perceive this as vague at the time of writing, we certainly want to clarify any statements that appear vague. It is not entirely clear here what references to published studies the reviewer refers to in this segment. We would be happy to make further amendments, if the statement is still not clear. For now, we will change it from:

"*Overall, the seismicity in the country is considered moderate with three to four earthquakes a day recorded on average within the country and around its borders by the Swiss Seismological Service (SED)*"

*to:*

"*Overall, the seismicity in the country is considered moderate with three to four earthquakes a day recorded on average within the country and around its borders by the monitoring network of the Swiss Seismological Service (SED; www.seismo.ethz.ch), the federal agency responsible for monitoring earthquakes in Switzerland and its neighboring regions.*"

7. Line 114: " site condition indicators... " which ones?

We have rephrased this sentence, hopefully our amendment has improved clarity. The sentence originally read:

"*The site condition indicators, including the lithological classification of Switzerland, multiscale topographic slope, and depth-to-bedrock, were combined with […]*".

We will change it to:

"*The selected site condition indicators (lithological classification of Switzerland, multiscale topographic slope, and depth-to-bedrock), were combined with […]*".

8. "Line 161: "...collected from other sources..." which ones?

We will add the following references:

Mouyiannou et al., 2014; Ottonelli et al., 2020; Rossi et al., 2021; Cardone, 2016; Cardone and Perrone, 2015; Magenes and Calvi, 1997; Avila et al., 2012.

9. Figure 1 is not cited in the body of the manuscript

Thank you for noticing it, we will cite it within the text.

10. Line 162: please specific the macroeconomic model. Do you think that structural/nonstructural damage-to-loss ratio from US can be transferred to Switzerland? In particular, this ratio could have been explored in greater depth with a few tests, so as to add something new to the study

We will largely expand this section by adding an additional page on the damage-to-loss ratio model. The full new section is given below:

[revised manuscript text omitted]

All in all, I think the paper could be eventually accepted if the issues set out above were strengthened and each step specified more precisely.